# ProTrain: Efficient LLM Training via Automatic Memory Management

## Abstract

Training billion-scale large language models (LLMs) with just a few consumer-grade graphics cards is key to democratizing LLM access. However, existing frameworks often depend on manual tuning of memory management settings, leading to inefficient hardware utilization and suboptimal performance. This paper introduces ProTrain, a novel training system that automatically tailors memory management policies to the model architecture and underlying hardware resources, eliminating the need for manual intervention. ProTrain features (1) automated memory management that abstracts complex memory management strategies into a few tunable configuration parameters and searches for optimal parameter settings using cost models and (2) a runtime profiler that provides precise estimates of latency, memory usage, and I/O bandwidth to build high-fidelity cost models. ProTrain does not change the training algorithm and thus does not compromise accuracy. Experiments show that ProTrain improves training throughput by $1.43\times$ to $2.71\times$ compared to the state-of-the-art training systems.

## 1 Introduction

Large Language Models (LLMs) have recently achieved remarkable success in various fields. Inspired by the scaling law Kaplan et al. (2020) that the performance (e.g., perplexity) of LLMs often improves logarithmically with the number of parameters, there has been a trend towards increasing parameter size. For instance, the parameter size of GPT-like models has surged from 117 million in GPT-1 Han et al. (2021) to 1,760 billion in GPT-4 Achiam et al. (2023), a 15,000-fold increase over two years. The significant growth in parameter size leads to a substantial increase in memory demands. According to existing studies Ren et al. (2021), each unit increase in parameters generally requires $16\times$ more memory to store the model states (e.g., fp16 and fp32 parameters, fp16 gradients, fp32 momentum and variances), not to mention the increased memory demand for activations due to larger model sizes. Consequently, memory has become the dominant bottleneck in LLM training.

Numerous memory management strategies have been proposed to address memory limitations. They generally fall into three categories: ZeRO, gradient checkpointing, and swapping. (a) The Zero Redundancy Optimizer (*ZeRO*) Rajbhandari et al. (2020); Zhao et al. (2023b) distributes model states across multiple GPUs, leverageing aggregated memory capacity to accommodate large models in data parallelism. (b) *Gradient checkpointing* Chen et al. (2016); Jain et al. (2020); Herrmann et al. (2019); Zhao et al. (2023a); Korthikanti et al. (2023) reduces memory consumption by discarding certain activations during the forward pass and recomputing them during the backward pass. (c) *Swapping* Rhu et al. (2016); Wang et al. (2018); Le et al. (2018); Huang et al. (2020); Ren et al. (2021); Rajbhandari et al. (2021); Sun et al. (2022) offloads data to external memory sources such as CPU memory or NVMe devices. As we consider swapping to CPU memory, we use *swapping* and *CPU offloading* interchangeably.

The three memory management strategies can be implemented within various model training paradigms, including data parallelism Ren et al. (2021), tensor parallelism Shoeybi et al. (2019), and pipeline parallelism Huang et al. (2019); Narayanan et al. (2019). This paper focuses on data parallelism, as it is widely used in distributed environments due to its simplicity and scalability.

Popular data-parallel frameworks, such as DeepSpeed Rasley et al. (2020), Colossal-AI Li et al. (2023)[1], and FSDP Zhao et al. (2023b), incorporate the aforementioned memory management strategies. However, these frameworks share a common issue: *they demand significant manual effort to configure memory management settings.* For example, in DeepSpeed, users must select the appropriate ZeRO optimization stage (e.g., ZeRO-1, ZeRO-2, ZeRO-3), configure offloading options (CPU or NVMe) for both parameters and optimizer states, and set various thresholds for parameter fetching and collective communications. Similarly, while Colossal-AI dynamically manages memory by moving data between the CPU and GPU, users must specify the non-model data ratio. The optimal configuration varies across models and hardware, requiring substantial domain expertise. Misconfiguration of these settings can lead to reduced efficiency or out-of-memory (OOM) error. For instance, GPT-10B running on four RTX 3090 GPUs with the default configuration utilizes only 35.6% of GPU memory and runs $1.18\times$ slower than with optimized settings. Moreover, configurations optimized for A100 GPUs cannot be directly applied to RTX 3090 GPUs due to high OOM risks.

To address this challenge, we propose ProTrain, an efficient LLM training system that automatically identifies memory management policies tailored to the specific LLM architecture and available memory resources. The basic idea of ProTrain is to abstract memory management strategies into a few tunable configuration parameters. ProTrain then builds runtime and memory usage estimators that quantify the impacts of these configuration parameters on training performance. These cost models, informed with accurate profiling information on latency, memory, and I/O bandwidth, allow ProTrain to search for the optimal memory management strategy that minimizes runtime while ensuring the peak memory consumption meets the hardware constraints.

Our main contributions are:

- *Automatic Memory Management* – To manage model states, we propose a dual-chunk system that treats initial layers of the LLM as persistent chunks in GPU memory and efficiently prefetches or offloads later layers as non-persistent chunks in CPU memory. For activation management, we introduce an interleaved organization that alternates between swapping and gradient checkpointing for each transformer block of the LLM. These strategies are abstracted into tunable configuration parameters, creating a structured configuration space that enables precise estimation of runtime and memory usage and facilitating the automatic search for optimal configurations with cost models.
- *Memory-Aware Runtime Profiling* – We are the first to apply model-wise runtime profiling to LLMs, leveraging detailed memory usage characteristics to reduce overall memory consumption. Building on this, we propose a novel memory-aware profiling method that effectively captures the memory consumption from temporary tensors often overlooked by state-of-the-art approaches, providing precise memory usage estimation to guide automated memory management.
- *Implementation of ProTrain* – We implement these techniques into a training system ProTrain that **automatically** configures memory management strategies, including CPU offloading, gradient checkpointing, and ZeRO techniques, to optimize training throughput while adhering to memory constraints.
- *Evaluation* – We ran ProTrain and other popular training frameworks (e.g., DeepSpeed, Colossal-AI, FSDP) on various models such as GPT-2, OPT, Mistral, and LLaMA. On RTX 3090 GPUs, ProTrain trained models up to $2.47\times$ larger than DeepSpeed and $1.48\times$ larger than Colossal-AI. On A100 GPUs, ProTrain trained models up to $7.5\times$ larger than FSDP, with $1.43\times$ to $2.71\times$ higher throughput than other frameworks. ProTrain also demonstrated excellent scalability with increasing GPUs or batch sizes. These results highlight ProTrain's superior memory management and efficiency across different hardware setups, making it an excellent choice for LLM training on memory-constrained settings.

## 2 BACKGROUND AND RELATED WORKS

This section introduces the background on DNN training. The discussion on more related works is in Appendix E.

---

[1] which rewrote PatrickStar Fang et al. (2022), and the two are used interchangeably in this paper

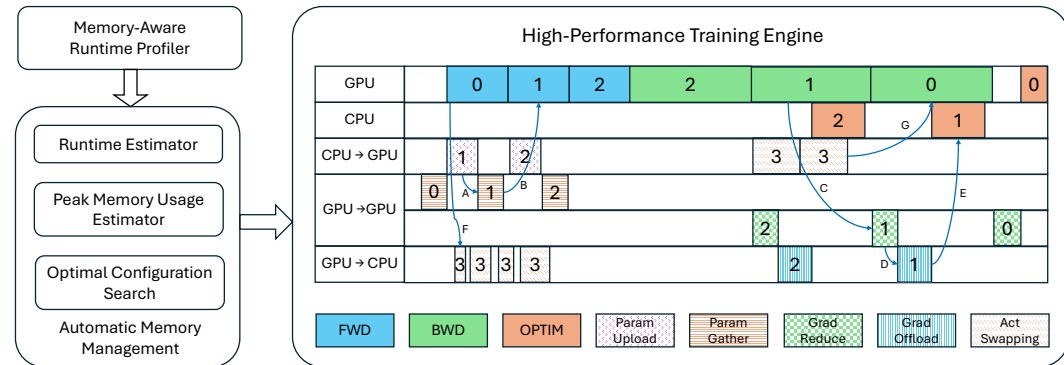

Figure 1: The architecture overview of ProTrain. Operations A - G are described in the text.

Training deep learning models involves a repetitive three-stage process across multiple iterations and epochs. The stages include forward propagation (FWD), where a batch of training samples is passed to the model to compute the loss; backward propagation (BWD), which calculates gradients by backpropagating the loss through the model; and parameter updating (OPTIM), where the gradients are used to update model parameters via an optimizer. For the training of large models, it is a common practice to adopt mixed-precision training Micikevicius et al. (2017), which uses reduced precision data types for FWD and BWD, while maintaining higher precision for OPTIM to ensure accuracy.

Memory consumption during training primarily comes from two sources: *model states* and *residual states*. Model states include parameters, gradients, and optimizer states (i.e. momentum and variances used in Adam Kingma & Ba (2014)) while residual states consist of activations and temporary tensors. The computational complexity of the FWD and BWD stages scales with model size and batch size, necessitating their execution on GPUs due to the intensive computational demands. In contrast, the OPTIM stage involves simpler operations and can be efficiently offloaded to the CPU Ren et al. (2021), which brings significant GPU memory savings by allocating memory-intensive optimizer states on the CPU.

## 3 OVERVIEW OF PROTRAIN

Figure 1 illustrates the system architecture of ProTrain, consisting of three core components: (1) an Automatic Memory Management module (§ 3.1) that automatically identifies the optimal memory management policy for training the target LLM on the given hardware, (2) a Memory-Aware Runtime Profiler (§ 3.2) that gathers runtime and memory data to guide memory management decisions, (3) a High-Performance Training Engine (§ 3.3) that implements the memory management policy. Before diving into each subsection, we first elaborate an example of memory management policies.

**Running Example.** The training engine diagram illustrates a memory management policy that Automatic Memory Management would discover. In the example, the LLM architecture is divided into three chunks, where each chunk represents a few consecutive transformer blocks. GPU performs the FWD, BWD, and a portion of OPTIM computations while CPU performs the rest of the OPTM computations. Since the parameters of Chunk 0 will be used immediately at the start of a training iteration, they are persistently allocated on the GPU. The parameters for Chunk 1 and Chunk 2 reside on the CPU and are dynamically uploaded to the GPU or offloaded back to the CPU to ensure the total memory consumption meets the device memory limit. The flow of communication operations between the CPU and GPUs is as follows:

(A) *Parameter Upload*: Before the forward pass, the parameters for Chunk 1 are uploaded from the CPU to the GPU. Since Chunk 0 already resides on the GPU, only Chunk 1 and Chunk 2 need to be uploaded sequentially from the CPU, illustrated as blocks 1 and 2 in the row "CPU → GPU". The prefetch of the next parameter chunk begins as soon as the GPU starts computing the current chunk.

(B) *Parameter Gather*: Once the parameter chunks are uploaded, the engine performs an *all-gather* operation that collects the parameter shards from all GPUs into a complete parameter chunk for

upcoming computations. This step is required for all three chunks, illustrated as blocks 0, 1, and 2 in the top row of "GPU → GPU".

(C) *Gradient Reduce*: In the backward pass, the engine reuses the parameter chunk to store the computed gradient to optimize memory usage. Once all parameters within a chunk are replaced by their corresponding parameters, a *reduce-scatter* operation is performed to synchronize gradients across GPUs, illustrated as blocks 0, 1, and 2 in the bottom row of "GPU → GPU".

(D) *Gradient Offload*: Following the gradient reduce, the chunks that were originally on the CPU, are offloaded back to the CPU to free up GPU memory. Only Chunk 1 and Chunk 2 perform this step.

(E) *Parameter Update*: Once on the CPU, the gradients for Chunk 1 and Chunk 2 are used for parameter updates, along with the high-precision parameter chunk already resided on the CPU. This step runs in parallel with the GPU's backward execution. In contrast, Chunk 0 performs its parameter updates directly on the GPU.

(F) *Activation Swapping Out*: Activation swapping occurs at the transformer block level, which is more fine-grained than chunks. In the example, only activations from the first transformer block (denoted as four block 3 in the row "GPU → CPU") are swapped out after each activation is computed.

(G) *Activation Swapping In*: When sufficient GPU memory is available to hold a transformer block's activations, the swapping in begins. This is done in batches (denoted as two block 3 in the row "CPU → GPU") rather than individually as swapping out shows, grouping multiple activations to improve bandwidth utilization.

In this example, parameter uploads from the CPU only occur during the forward pass assuming the GPU has enough buffer capacity to hold all the parameter chunks. However, if the buffers become full, the least recently used chunk is evicted, triggering another parameter upload and gather operation during the backward pass. Throughout the process, communication overhead is minimized by overlapping data transfers with computations. Additionally, idle CPU cycles are used to perform parameter updates, which run concurrently with the GPU's backward computations, effectively hiding slower CPU parameter update operations.

## 3.1 AUTOMATIC MEMORY MANAGEMENT

The Automatic Memory Management module abstracts the memory management policy into a few configuration parameters and automatically tunes these parameters to optimize the training efficiency of a LLM on a target hardware. We next elaborate on the abstractions of the configuration space and the optimal configuration search algorithm.

### 3.1.1 THE CONFIGURATION SPACE OF MEMORY MANAGEMENT

**Configuration Parameters for Model States.** Model states can be offloaded to the CPU to relieve GPU memory pressure but the offloading implementations in existing LLM training frameworks suffer from various limitations. Fully offloading all parameters, as seen in FSDP Zhao et al. (2023b), often leads to inefficient GPU memory usage and high data transfer overhead. DeepSpeed Rasley et al. (2020) attempts to mitigate this issue by using thresholds, such as maximum live parameters and prefetch bucket size, to control the offloading ratio. However, its prefetching mechanism operates in a sliding window manner due to poorly timed execution, resulting in frequent small transfers. This causes low bandwidth utilization, significantly degrading performance. Colossal-AI Li et al. (2023) improves bandwidth utilization through fixed-sized chunks but suffers from frequent memory reallocations caused by dynamic chunk management. Moreover, it uploads high-precision parameter chunks for GPU parameter updates at runtime, increasing the risk of memory fragmentation and out-of-memory (OOM) errors.

To address these limitations, ProTrain introduces a **dual-chunk system** consisting of persistent and non-persistent chunks. Persistent chunks remain on the GPU, storing both high-precision and low-precision parameters, which eliminates data transfers and enables direct GPU parameter updates. In contrast, non-persistent chunks are kept in CPU memory, requiring low-precision parameters uploads to the GPU for computation, and gradients offloads back to the CPU for parameter updates. For non-persistent chunks, ProTrain further introduces **pre-allocated chunk buffers** that are used as caches. These buffers allow parameters loaded during the forward pass to be reused in the backward

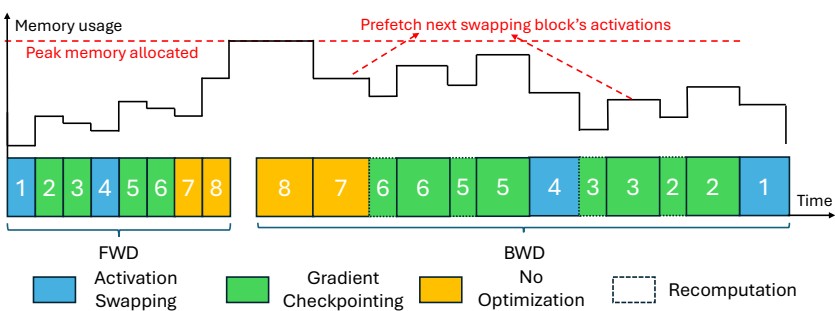

Figure 2: Block-Wise Activation Management Layout and Memory Usage Trend

pass, preventing frequent memory allocations. In ProTrain, persistent chunks are the first few chunks of the LLM while the non-persistent chunks are the rest of the chunks.

The concept of dual-chunk system allows ProTrain to tailor offloading policies to chunks of different characteristics. Dual-chunk system is inspired by two observations: (1) The forward pass computation can often hide the overhead of CPU offloading for the later layers of a LLM but not that of the first few layers. (2) The parameter updates for the later layers of a LLM, but not the first few layers, can be performed concurrently with the backward pass computation. Therefore, *the model states of the first few layers and the later layers should be managed differently*.

We use the example in Figure 1 to explain the rationale. As chunk 0 executes first in the forward pass, if its parameters are offloaded to the CPU, its forward pass computation will be blocked by the data transfer overhead from CPU-to-GPU parameter uploading and parameter gather. Chunk 0 also gets updated the last with no backward pass computation left to hide the latency from parameter updates. Therefore, managing Chunk 0 as a persistent chunk eliminates cold start latency and enables efficient GPU parameter updates. In contrast, the data transfer necessary for Chunk 1 and Chunk 2 to perform forward pass can be overlapped with the computation of Chunk 0 and Chunk 1 respectively. If performed on the CPU, the parameter updates of Chunk 1 and Chunk 2 can also be overlapped with the backward pass computation of Chunk 0 and Chunk 1 respectively. Therefore, managing Chunk 1 and Chunk 2 as non-persistent chunks relieves GPU memory pressure without incurring offloading overheads.

We summarize the configuration parameters from managing model states as follows: (1) **chunk size** – the size of each chunk for the LLM, (2) **the number of persistent chunks**, and (3) **the size of pre-allocated chunk buffers**. In particular, while more persistent chunks and chunk buffers generally improve performance, memory constraints and the large size of LLMs necessitate a trade-off between memory usage and system efficiency.

**Configuration Parameters for Activations.** Previous studies Peng et al. (2020); Beaumont et al. (2021) have co-optimized activation swapping and gradient checkpointing at the tensor granularity. Although tensor-level management offers greater flexibility, it significantly expands the search space, making it challenging to determine optimal policies for swapping or recomputing individual tensors. For instance, the LLaMA 34B model has only 48 transformer blocks but has approximately 2,000 activation tensors, resulting in a search space as large as $3^{2000}$ if each tensor has three options. Moreover, managing tensors individually introduces implementation complexities and scalability challenges, making this approach impractical for LLMs. In contrast, popular training frameworks that utilize gradient checkpointing often recompute all transformer blocks, which is inefficient when there is sufficient memory to avoid full recomputation.

To address the above limitations, ProTrain takes a different approach by managing activation swapping and gradient checkpointing operations **at the transformer block level**. Each block can utilize one of three techniques in handling activations: swapping, gradient checkpointing, or no optimization (i.e., neither swapping nor checkpointing is applied). To enhance efficiency, ProTrain introduces an **interleaved organization**, in which each swapping block is followed by multiple blocks using gradient checkpointing. This design offers several benefits. First, placing swapping blocks earlier increases opportunities for overlapping swapping with computation. Second, interleaving them with checkpointing blocks prevents activation accumulation, reducing the risk of OOM errors caused by

slower swapping. Third, placing unoptimized blocks in the later layers allows their activations to be consumed sooner, enabling earlier activation prefetching of swapping blocks.

Figure 2 illustrates our approach using a transformer with 8 blocks. Block 1 and 4 use swapping, while block 2, 3, 5, and 6 use gradient checkpointing. The remaining blocks are left unoptimized, as their earlier backward computations offer limited opportunities for swapping. This interleaved approach not only maximizes the overlap between computation and communication, but also minimizes peak memory usage, as visualized in the upper part of Figure 2.

We summarize the configurable parameters from managing activations as follows: (1) **the swapping interval**, which is selected based on the computation time needed to swap out a block, (2) **the number of blocks designated for swapping and gradient checkpointing**. Striking a balance between the number of swapping and checkpointing blocks is crucial: ideally, fewer blocks should use either technique, as each introduces additional recomputation or transfer overhead. However, when memory is constrained, swapping is preferred for blocks where communication overhead can be effectively hidden.

### 3.1.2 OPTIMAL CONFIGURATION SEARCH WITH COST MODELS

We formulate the optimal configuration search as a constrained optimization problem. The goal is to minimize the total runtime of the training process. Since training consists of repeated iterations, minimizing the total training time is equivalent to minimizing the runtime of a single iteration, denoted as $T_{\text{Iteration}}$, which includes the forward pass, backward pass, and parameter updates:

$$\min_{configs} T_{\text{Iteration}} \quad s.t. \ M_{\text{Peak}} < M_{\text{Capacity}}, \tag{1}$$

where $M_{\text{Peak}}$ represents the peak memory usage, and $M_{\text{Capacity}}$ is the total GPU memory capacity. The set of tunable configuration parameters, *configs*, that determines the memory management policy is $configs = \{n_{\text{persist}}, n_{\text{buffer}}, n_{\text{swap}}, n_{\text{checkpoint}}\}$, where $n_{\text{persist}}$ denotes the number of persistent chunks residing on the GPU, $n_{\text{buffer}}$ refers to the number of chunk buffers for prefetching and memory reuse, $n_{\text{swap}}$ indicates the number of blocks using activation swapping, and $n_{\text{checkpoint}}$ specifies the blocks applying gradient checkpointing. These configurations are non-negative integers that are bounded by the total number of chunks ($N_{\text{chunk}}$) or blocks ($N_{\text{block}}$). Chunk size is determined independently before the optimal configuration search (detailed in Appendix B.1).

To solve the optimization problem, we build two cost models that accurately estimate runtime and peak memory consumption for each configuration combination. These cost models allow us to identify the optimal configuration setting leveraging profiling information data alone, getting rid of the tedious trial-and-errors to set up training processes. The profiler is discussed in § 3.2.

**Runtime Estimator.** In ProTrain, CPU parameter updates are executed concurrently with the GPU's computations, which include both the backward pass and GPU-based parameter updates. However, if the CPU parameter updates cannot fully overlap with the GPU's operations, the total iteration time becomes constrained by the longer CPU update phase. The runtime cost model is formulated as:

$$T_{\text{Iteration}} = T_{\text{FWD}} + max\{T_{\text{BWD}} + T_{\text{GPU\_OPTIM}}, T_{\text{CPU\_OPTIM}}\}, \tag{2}$$

where $T_{\text{FWD}}$ and $T_{\text{BWD}}$ are modeled as a function of the configuration parameters. For parameter update of the persistent chunks ($T_{\text{GPU\_OPTIM}}$) and non-persistent chunks ($T_{\text{CPU\_OPTIM}}$), ProTrain models runtimes predictably based on parameter size. Due to space limitations, details are in Appendix A.1.

**Peak Memory Usage Estimator.** Memory usage falls into two categories: static and dynamic components. ***Static memory***, which includes model states and activations, is fixed and predictable. They can be easily determined by chunk size, $n_{\text{persist}}$, and $n_{\text{buffer}}$. However, ***dynamic memory*** involves temporary tensors that are hard to estimate and are often neglected in existing approaches Wang et al. (2024); Huang et al. (2022). Although transient, these temporary tensors can significantly impact peak memory usage, accounting for up to 17.2% (3.06 GB) of total memory. To address this, we design an iterative operator-wise approach to estimate peak memory usage. The basic idea is to track peak memory during profiling while excluding static memory, then iteratively add back the static memory during the estimation phase, operator by operator, to accurately capture the contribution of temporary tensors to the overall peak memory. Details of the algorithm are given in the Appendix A.2.

The configuration space in ProTrain is structured and finite, allowing for an exhaustive search of all possible configurations. ProTrain employs specific pruning strategies to further reduce the

search space. For instance, the maximum number of swappable blocks is limited by the swapping interval to ensure they overlap with forward computations. During the backward phase, the system monitors bandwidth usage for chunk prefetching to ensure sufficient bandwidth remains for activation prefetching. Additionally, as configurations are traversed from smallest to largest, any swapping and checkpointing combination that results in memory overflow is immediately discarded, and subsequent iterations involving this combination are skipped. For each viable configuration, ProTrain's runtime estimator predicts the runtime, selecting the one with the shortest runtime as the final setup.

### 3.2 MEMORY-AWARE RUNTIME PROFILING

Traditional memory profiling methods, such as static profiling Patil et al. (2022) and layer-wise runtime profiling Beaumont et al. (2021), are insufficient for capturing the complete memory demands of LLM training. These approaches often overlook the impact of unhookable operators and temporary tensors, leading to inaccurate memory management and suboptimal configuration choices. Model-wise runtime profiling has the potential to overcome these challenges. However, as it requires the execution of the entire LLM model, it is constrained by limited GPU memory capacity for LLMs.

ProTrain develops an memory-aware runtime profiling system that leverages memory usage characteristics to enable model-wise profiling with limited memory capacity. Specifically, ProTrain drops static memory (e.g., parameters, gradients, activations) from the GPU and regenerates it when required. This is based on the observation that static memory usage is predictable (as detailed in Section 3.1.2), allowing the profiler to focus on capturing the more complex and transient dynamic memory usage.

To track dynamic memory fluctuations caused by temporary tensors and unhookable operators, ProTrain registers hooks that monitor current and peak memory changes both before and during operations. First, the peak memory usage during each operation is monitored to capture the temporary tensor usage specific to that operation. Second, by analyzing the memory changes between consecutive hookable operations, the profiler infers the memory usage of unhookable operators. This operator-wise approach considers the life cycle of various tensors, enabling a more precise understanding of memory usage dynamics and making the profiler memory-aware, which is crucial for building accurate cost models.

Our profiler also tracks the execution time of each operator. Similar to memory profiling, we estimate the execution times of unhookable operators by analyzing the intervals between hookable ones. Additionally, the profiler collects detailed hardware metrics, including memory transfer bandwidth and collective communication operation durations, under both isolated and overlapping scenarios. This detailed data collection enables precise performance predictions and facilitates automatic memory management tailored to specific models and hardware, as discussed in Appendix A.

### 3.3 HIGH-PERFORMANCE TRAINING ENGINE IMPLEMENTATION

ProTrain is implemented on top of PyTorch, with a total of 7,600 lines of code. It offers simple and user-friendly APIs, which require less than 5 lines of code modification to integrate with existing PyTorch training scripts. Unlike existing approaches Rasley et al. (2020); Li et al. (2023), ProTrain eliminates the need for manual configuration through its automatic memory management system. ProTrain also includes several memory optimization techniques, detailed in Appendix B.2.

## 4 EXPERIMENTS

We empirically evaluate the performance of ProTrain against three open-source LLM training frameworks using four popular LLM architectures.

**Workloads.** The tested models includes GPT-2 Radford et al. (2019), OPT Zhang et al. (2022), Mistral Jiang et al. (2023), and LLaMA Touvron et al. (2023). By varying the hidden dimension, the number of transformer blocks, and the number of attention heads, we generate models with different parameter sizes, detailed in the Appendix C.1. The sequence length is set to 1024 by default.

**Testbed.** We evaluate the performance of ProTrain in two different experimental environments: (1) 1 node of 4 NVIDIA GeForce RTX 3090 24GB with 384GB of DRAM; (2) 1 node of 4 NVIDIA A100 SXM4 80GB with NVLink 3.0 with 1TB of DRAM. Details are provided in Appendix C.2.

**Baselines.** We compare ProTrain with three representative open-source LLM training solutions: (1) **FSDP** Zhao et al. (2023b), the native PyTorch support for the ZeRO-3 technique; (2) **DeepSpeed** Rasley et al. (2020), a widely-used distributed training framework that employs ZeRO and offloading techniques, tested with ZeRO-3 for a fair comparison; and (3) **Colossal-AI** Li et al. (2023), which adopts chunk-based memory management compatible with the ZeRO-3 technique. Details on baseline configurations are provided in Appendix C.3.

## 4.1 TRAINING PERFORMANCE COMPARISON

**Maximum Trainable Model Size.**
Table 1 reports the maximum trainable model sizes for different frameworks, using the GPT-2 model as the benchmark. ProTrain demonstrates superior performance, supporting models up to 34 billion parameters on a single RTX 3090 GPU and scaling to 37 billion with four GPUs. On the more powerful A100

Table 1: Maximum Trainable Model Size (Unit: Billion)

| Backend | RTX 3090*1 | RTX 3090*4 | A100*1 | A100*4 |
|---|---|---|---|---|
| ProTrain | **34B** | **37B** | **75B** | **87B** |
| DeepSpeed | 15B | 15B | 34B | 37B |
| Colossal-AI | 25B | 25B | 53B | 53B |
| FSDP | 1B | 15B | 10B | 55B |

GPU, ProTrain trains models as large as 75 billion on a single GPU and 87 billion with four GPUs, outperforming Colossal-AI and DeepSpeed by 1.64× and 2.35×, respectively, in the four-GPU setup. In contrast, FSDP significantly underperforms in the single GPU setting, managing only much smaller models compared to ProTrain. Some frameworks fail to scale model sizes with more GPUs, primarily due to inefficiencies in handling model initialization across devices. These results highlight ProTrain's effective utilization of heterogeneous memory resources, democratizing the LLM training.

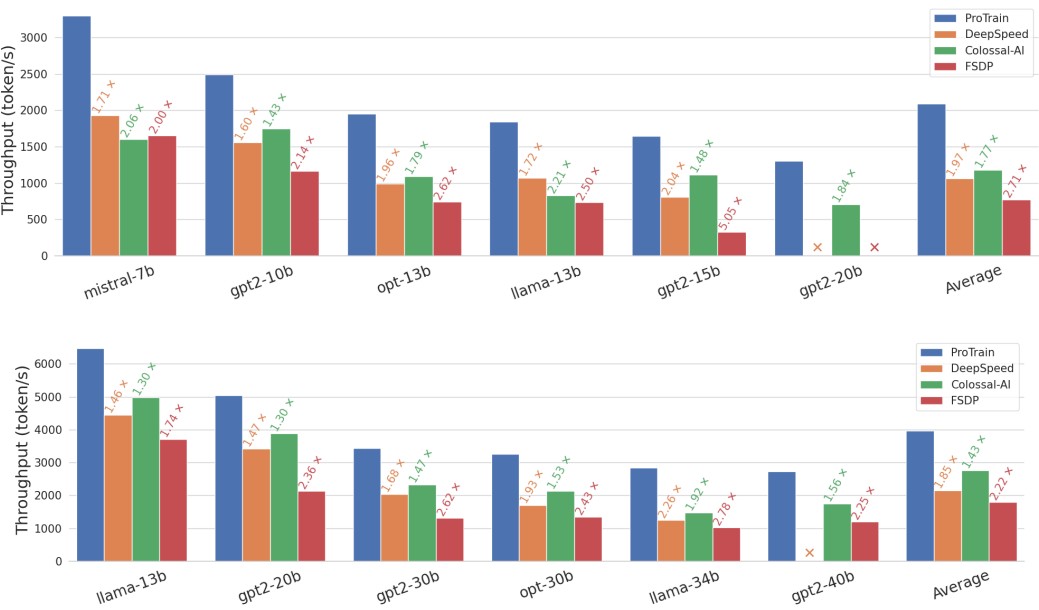

Figure 3: Maximum Training Throughput on four RTX 3090 GPUs (upper) and A100 GPUs (bottom). The notation "×" indicates failure to train due to out of memory.

**Training Throughput.** Figure 3 presents the maximum training throughput for various models on four RTX 3090 and A100 GPUs, measured in tokens per second. The throughput is obtained by testing each model at different batch sizes to find the highest achievable throughput. The results show that ProTrain consistently outperforms other frameworks across diverse hardware and models. On RTX 3090 GPUs, ProTrain achieves an average throughput of 2089.50 tokens per second, 1.77 to 2.71× higher than other frameworks. On A100 GPUs, ProTrain improves the throughput of DeepSpeed, Colossal-AI, and FSDP by 1.85×, 1.43×, and 2.22×, respectively.

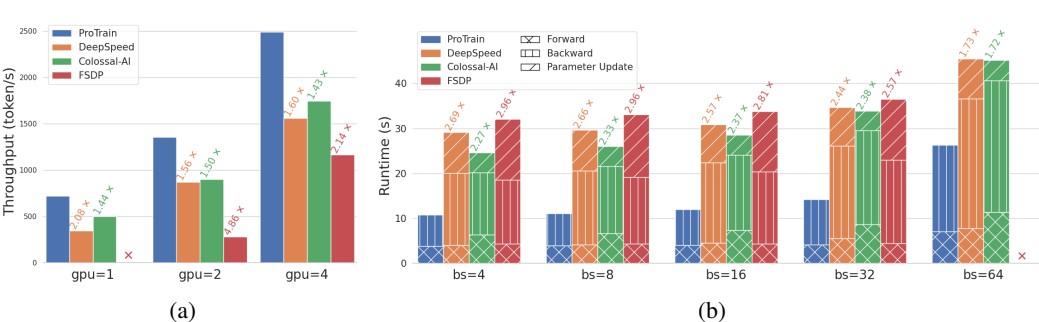

Figure 4: Scalability of performance on RTX 3090 GPUs (a) Maximum throughput across different numbers of GPUs (b) Step time breakdown for different batch sizes

As model sizes increase, the demand for memory resources grows, resulting in decreased training performance. However, ProTrain consistently maintains robust performance compared to other frameworks. Notably, ProTrain delivers substantial speedups, achieving $5.05\times$ the training speed of 15B GPT-2 on RTX 3090 and $2.78\times$ of 34B LLaMA on A100, compared to FSDP. In such cases, other frameworks either fail to train larger models with feasible batch sizes or resort to inefficient data offloading. Overall, ProTrain delivers substantial performance improvements, achieving up to $2.71\times$ the throughput of other frameworks on average, significantly enhancing the efficiency of LLM training.

**Performance Scalability.** Figure 4(a) shows the maximum throughput of 10B GPT-2 across varying GPU counts. ProTrain demonstrates impressive scalability, reaching 2493 token/s with four GPUs, a $3.5\times$ increase from a single GPU setup. In contrast, while DeepSpeed and Colossal-AI also increase throughput with more GPUs, their performance gains do not match those of ProTrain.

**Performance Breakdown.** Figure 4(b) provides a detailed breakdown of iteration time into forward, backward, and parameter update phases when training a 10B GPT-2 model at varying batch sizes on four RTX 3090 GPUs. At smaller batch sizes, where GPU memory pressure is lower, ProTrain significantly outperforms other frameworks for two reasons. First, ProTrain optimizes both computations and I/O through overlapping, effectively hiding much of the latency. This is evident from the figure, where ProTrain's parameter update time is nearly negligible compared to other phases, due to its efficient overlap with backward computations. Second, ProTrain's automatic memory management module dynamically identifies the optimal balance of memory-saving techniques, improving both memory efficiency and performance. As batch sizes increase, the runtime for one iteration generally rises across all frameworks due to heavier computational and memory demands. In these cases, ProTrain maximizes memory-saving techniques, with performance gains primarily driven by better overlapping strategies. Appendix D.1 presents experimental results on A100 GPUs.

## 4.2 ABLATION STUDIES

**Importance of the Configuration Parameters.** Figure 5(a) illustrates the impact of removing key optimization components in ProTrain when training a 10B GPT-2 model on four RTX 3090 GPUs. Without *dual-chunk system*, where persistent chunks are replaced by three chunk buffers, we observe a $1.1\times$ slowdown. As batch sizes grow and memory pressure increases, the optimal configuration shifts toward fewer persistent chunks and chunk buffers, limiting further speedup. However, ProTrain automatically adapts its memory management to match the model architecture and hardware conditions, ensuring efficient resource utilization across various workloads. Similarly, disabling the *interleaved organization* and applying gradient checkpointing to all transformer blocks results in an average $1.04\times$ slowdown. While the benefit of the interleaved organization diminishes at larger batch sizes, ProTrain dynamically adjusts the number of blocks for swapping and checkpointing to strike the optimal balance between memory efficiency and computational overhead. The largest performance degradation occurs when the *overlapped parameter update* is removed. Switching to a sequential approach results in a $1.22\times$ slowdown. This aligns with Figure 4(b), where ProTrain's optimized parameter update greatly reduces its share of the overall runtime. Appendix D.5 summarizes the combinations of techniques that achieve optimal memory management and performance.

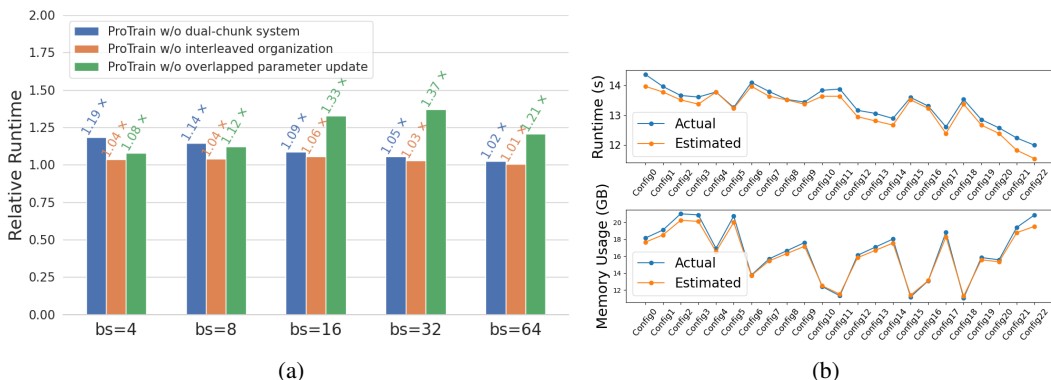

(a)  (b)

Figure 5: (a) Effectiveness of dual-chunk system, interleaved organization, and overlapped parameter update. The speedup on each bar reports the time spent by ProTrain w/o the optimization divided by the time spent by ProTrain. (b) Effectiveness of runtime and peak memory usage estimator.

**Effectiveness of Runtime Estimator.** The upper chart in Figure 5(b) demonstrates the effectiveness of the runtime estimator by comparing the estimated and actual runtimes for various configurations during the training of the 10B GPT-2 model. The estimator consistently provides accurate predictions, with the gaps staying within 4% across a wide range of configurations. This highlights its robustness in managing diverse memory optimization strategies. We also confirm the generalizability of the runtime estimator across different models and hardware setups. With precise runtime estimates, ProTrain can automatically determine the most efficient memory management configurations for specific models and hardware.

**Effectiveness of Peak Memory Usage Estimator.** We demonstrate that the estimated memory usage is within 7% error of actual usage, as shown in the bottom chart of Figure 5(b). This high accuracy ensures that the optimal configurations identified by the runtime estimator are not only efficient but also safe, effectively preventing the risk of OOM errors during training. Appendix D.4 further shows the predicted and actual runtime and peak memory usage for various models and batch sizes.

## 5 DISCUSSION

ProTrain is designed for small clusters, which may pose challenges in large-scale training where cross-GPU communication overhead becomes more significant. However, in our preliminary experiments, where we trained a 15B GPT-2 model with a batch size of 160 across two nodes (each equipped with four V100 GPUs), ProTrain showed promising results, outperforming DeepSpeed by $1.53\times$ and Colossal-AI by $1.84\times$, while FSDP encountered OOM errors. Notably, these results were achieved without any dedicated optimizations for multi-node environments in ProTrain, highlighting the potential for further refinement and performance improvements.

Furthermore, ProTrain's ability to independently profile each node makes it well-suited for adapting to heterogeneous setups, opening up opportunities to explore optimizations across diverse hardware configurations. As future work, we aim to enhance ProTrain's performance in large-scale, multi-node environments by leveraging these optimization opportunities.

## 6 CONCLUSION

This paper introduced ProTrain, a novel training system designed to simplify the training process through automatic memory management. ProTrain highlights the significance of precise memory usage and runtime data gathered through memory-aware, model-wise profiling to build high-fidelity cost models, along with the careful abstraction of configuration parameters from memory management strategies to automate optimal configuration search. ProTrain achieves up to $5\times$ the performance of existing state-of-the-art frameworks and enables the training of models with up to 75 billion parameters on a single A100 GPU. We hope our work helps AI researchers and practitioners with limited GPU resources, making LLMs more accessible to a wider audience.

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

# A  COST MODELS

## A.1  MODELING RUNTIME

The total iteration time in ProTrain is determined by the duration of the forward pass, backward pass, and parameter updates, as defined in Equation 2. To estimate the forward computation time, ProTrain adopts a chunk-based approach, as most operations in Figure 1 operate at the chunk level. By comparing the computation and communication overheads for each chunk, the estimator identifies whether the chunk is compute-bound or communication-bound, using the larger value as its runtime estimate:

$$T_{\text{FWD}} = \sum_{i=1}^{N_{\text{chunk}}+1} \max\left(T_{\text{comp}}^{\text{FWD}}(i-1), T_{\text{comm}}^{\text{FWD\_prefetch}}(i)\right), \tag{3}$$

where $T_{\text{comp}}^{\text{FWD}}$ represents the forward computation time of a chunk, which aggregates the runtimes of individual operators within the chunk. $T_{\text{comm}}^{\text{FWD\_prefetch}}$ represents the communication time required to prefetch parameters for the next chunk during the forward pass, which is calculated as follows:

$$T_{\text{comm}}^{\text{FWD\_prefetch}}(i) = \begin{cases} T_{\text{comm}}^{\text{gather}}(i), & \text{if } i \leq n_{\text{persist}}, \\ 0, & \text{if } i > N_{\text{chunk}}, \\ T_{\text{comm}}^{\text{gather}}(i) + T_{\text{comm}}^{\text{upload}}(i), & \text{otherwise}, \end{cases} \tag{4}$$

where $T_{\text{comm}}^{\text{gather}}$ is the time to gather parameter chunks from multiple GPUs, and $T_{\text{comm}}^{\text{upload}}$ is the time to transfer non-persistent chunks from CPU to GPU. To estimate $T_{\text{comm}}^{\text{gather}}$ and $T_{\text{comm}}^{\text{upload}}$, ProTrain uses detailed profiling to accurately model their runtime. In contrast to conventional approaches that assume a fixed bandwidth for memory transfers, ProTrain simulates various overlapping scenarios to capture the effects of bandwidth contention. For instance, when activation swapping is enabled, we estimate the swapping time, identify the affected chunks, and use the reduced bandwidth instead. The activation swapping time is excluded from the forward pass calculation, as ProTrain carefully controls $n_{\text{swap}}$ to ensure its overhead is fully overlapped with computation.

Similarly, the runtime of the backward pass is calculated at the chunk level:

$$T_{\text{BWD}} = \sum_{i=1}^{N_{\text{chunk}}+1} \max\left(T_{\text{comp}}^{\text{BWD}}(i) + T_{\text{recomp}}(i), T_{\text{comm}}^{\text{BWD\_prefetch}}(i-1), T_{\text{comm}}^{\text{reduce-offload}}(i+1)\right). \tag{5}$$

In contrast to the forward pass, the backward computation includes additional recomputation overheads from gradient checkpointing, represented by $T_{\text{recomp}}(i)$. The value is calculated as the aggregated forward computation time for the checkpointed blocks within chunk $i$, following the block-to-chunk mapping in the interleaved organization. Another key distinction from the forward pass is the overhead related to gradient reduce and offloading during the backward pass, represented by $T_{\text{comm}}^{\text{reduce-offload}}$, which is defined as:

$$T_{\text{comm}}^{\text{reduce-offload}}(i) = \begin{cases} T_{\text{comm}}^{\text{reduce}}(i), & \text{if } i \leq n_{\text{persist}}, \\ 0, & \text{if } i > N_{\text{chunk}}, \\ T_{\text{comm}}^{\text{reduce}}(i) + T_{\text{comm}}^{\text{offload}}(i), & \text{otherwise}. \end{cases} \tag{6}$$

As with $T_{\text{comm}}^{\text{FWD\_prefetch}}$, the performance of $T_{\text{comm}}^{\text{reduce-offload}}$ is directly influenced by the number of persistent chunks, as persistent chunks avoid parameter prefetching and only involve gradient reduce. However, $T_{\text{comm}}^{\text{BWD\_prefetch}}$ differs in its estimation from $T_{\text{comm}}^{\text{FWD\_prefetch}}$, and is defined as:

$$T_{\text{comm}}^{\text{BWD\_prefetch}}(i) = \begin{cases} 0, & \text{if } i \leq n_{\text{persist}} \text{ or } i > N_{\text{chunk}} - n_{\text{buffer}}, \\ T_{\text{comm}}^{\text{gather}}(i) + T_{\text{comm}}^{\text{upload}}(i), & \text{otherwise}. \end{cases} \tag{7}$$

This difference arises because of the presence of chunk buffers, which cache the parameter loaded and gathered during the forward pass, eliminating the need for re-loading and re-gathering in the backward pass. As a result, uploading and gathering are only required for chunks that were evicted due to limited buffer capacity.

Following the backward pass, parameter updates are executed on both the GPU and CPU, depending on the chunk placement. For CPU-based updates, ProTrain employs the fast CPU Adam optimizer Ren et al. (2021), while GPU updates use the FusedAdam optimizer NVIDIA (2018). ProTrain models performance for both updates based on parameter size.

## A.2 Modeling Memory Consumption

Accurately estimating peak memory usage is essential for efficient memory management, particularly in LLMs, where memory constraints require careful data handling to prevent exceeding capacity. Our estimator relies on the data collected by the profiler (detailed in Section 3.2) to compute memory usage precisely. The profiled data includes the changes in current memory usage, $\Delta M_{\text{Cur}}^{\text{PriorOp}}$, and peak memory usage, $\Delta M_{\text{Peak}}^{\text{PriorOp}}$, before each operation, as well as $\Delta M_{\text{Cur}}^{\text{Op}}$ and $\Delta M_{\text{Peak}}^{\text{Op}}$ during each operation. Additionally, the profiler tracks the activation memory usage for each operator, $M_{\text{Act}}^{\text{Op}}$, and the memory usage at the end of the forward pass, $M_{\text{FWD}}$. Since memory usage typically peaks during the backward pass, our focus is on identifying the peak memory usage in that phase.

To estimate peak memory usage, we define two key variables: the current memory usage, $M_{\text{Cur}}$, and the peak memory usage, $M_{\text{Peak}}$. Initially, $M_{\text{Cur}}$ is set to $M_{\text{FWD}} + \sum_{i=1}^{N_{\text{op}}} M_{\text{Act}}^{\text{Op}}(i)$. These values are iteratively updated for each operator using Equation 8 and 9:

$$M_{\text{Cur}}(i) = M_{\text{Cur}}(i-1) + \Delta M_{\text{Cur}}^{\text{PriorOp}}(i) + \Delta M_{\text{Cur}}^{\text{Op}}(i) - M_{\text{Act}}^{\text{Op}}(i), \tag{8}$$

$$M_{\text{Peak}}(i) = max\{M_{\text{Peak}}(i-1), M_{\text{Cur}}(i-1) + \Delta M_{\text{Peak}}^{\text{PriorOp}}(i), \\ M_{\text{Cur}}(i-1) + \Delta M_{\text{Cur}}^{\text{PriorOp}}(i) + \Delta M_{\text{Peak}}^{\text{Op}}(i)\}. \tag{9}$$

This iterative, operator-wise approach allows us to recover the peak memory usage by accounting for both the transient nature of temporary tensors, which are typically confined to individual operators, and the longer life cycle of activations, which span across multiple operations depending on the execution order. The final value obtained from Equation 9, denoted as $M_{\text{Peak}}^{\text{Base}}$, serves as the foundational baseline for estimating peak memory usage across various configurations. Building on this, the final peak memory for any specific configuration is computed as:

$$M_{\text{Peak}} = M_{\text{Peak}}^{\text{Base}} + M_{\text{persist}} \cdot n_{\text{persist}} + M_{\text{buffer}} \cdot n_{\text{buffer}} - M_{\text{swap}} \cdot n_{\text{swap}} \\ -M_{\text{checkpoint}} \cdot n_{\text{checkpoint}} + \begin{cases} M_{\text{checkpoint}}, & \text{if } n_{\text{checkpoint}} + n_{\text{swap}} = N_{\text{block}}, \\ 0, & \text{otherwise}, \end{cases} \tag{10}$$

where $M_{\text{persist}}$ and $M_{\text{buffer}}$ represent the memory allocated for a single persistent chunk and chunk buffer, and $M_{\text{swap}}$ and $M_{\text{checkpoint}}$ reflect the memory savings from activation swapping and gradient checkpointing for a single transformer block, respectively. When all blocks are involved in either swapping or gradient checkpointing, recomputation during the backward pass is inevitable, leading to an increase in memory consumption. Furthermore, actual memory usage is typically higher than estimates due to memory fragmentation, so we include a fragmentation factor in the final estimation.

## B Implementation Details

### B.1 Adaptive Chunk Size

ProTrain employs a dynamic search mechanism to determine the optimal chunk size for model training, which organizes parameters according to their execution order and ensures that all parameters within a block are grouped in a single chunk. For transformers that share parameters across layers,

ProTrain uses the parameter's first occurrence as the ordering criterion. To find the most efficient chunk size, ProTrain conducts a grid search, simulating memory waste across various chunk sizes to identify the size that minimizes waste.

## B.2 MEMORY OPTIMIZATIONS

**Proactive Memory Allocation**   ProTrain preallocates memory for tensors that persist until training completes, including early allocation of persistent chunks for parameters and optimizer states, as well as GPU chunk buffers. This proactive strategy reduces the number of memory allocations and mitigates fragmentation by grouping long-lived tensors together, ensuring a more organized and efficient memory layout.

**Single-Stream Memory Allocation**   ProTrain unifies memory allocations within the default stream to improve memory utilization. PyTorch's allocator adopts a multi-heap design where each stream has its own heap, limiting cross-heap memory reuse and necessitating the use of `record_stream()` to ensure correctness. By using a single stream for all allocations and directly managing deallocation synchronization ourselves, we effectively prevent misuse and reallocation conflicts, thereby improving memory efficiency.

**Customized Pinned Memory Allocator**   We observe that the default pinned memory allocator (`CUDAHostAllocator`) often over-allocates by rounding up to the nearest power of two, leading to significant memory waste. To address this inefficiency, ProTrain developed a customized pinned memory allocator that leverages insights from automatic memory management to precisely determine pinned memory requirements, providing finer control and avoiding the excessive memory reservation of the default allocator.

## C EXPERIMENT SETTINGS

### C.1 MODEL CONFIGURATIONS

The model configurations used in the experiment are shown in Table 2. The underlying model implementation is from the HuggingFace library.

Table 2: Model Configuration

| Model | Parameter Size | Hidden Size | # of Layers | # of Heads |
|---|---|---|---|---|
| Mistral | 7B | 4096 | 32 | 32 |
| GPT-2 | 10B | 4096 | 48 | 32 |
| OPT, LLaMA | 13B | 5120 | 40 | 40 |
| GPT-2 | 15B, 20B, 30B, 40B | 8192 | 18, 24, 36, 50 | 64 |
| OPT | 30B | 7168 | 48 | 56 |
| LLaMA | 34B | 8192 | 48 | 64 |

### C.2 HARDWARE CONFIGURATIONS

**4× RTX 3090**: The system contains four NVIDIA GeForce RTX 3090 GPUs with 24GB memory. It is powered by Intel(R) Xeon(R) Silver 4214R CPU @ 2.40GHz with 24 cores. The CPU DRAM size is 384GB. The PCIe version is 3 with 15.8GB/s bandwidth. NVLink is not available in this setup.

**4× A100**: The system contains four NVIDIA A100 GPUs with 80GB memory. It is powered by Intel(R) Xeon(R) Platinum 8480+ with 112 cores. The CPU DRAM size is 1TB. The PCIe version is 4 with 31.5GB/s bandwidth. GPUs are fully connected by NVLink 3.0 with 300GB/s bandwidth.

### C.3 BASELINE CONFIGURATIONS

For our experiments, we used DeepSpeed-0.12.1 with ZeRO-3 enabled, including offloading of both parameters and optimizer states. Parameters and gradients were grouped at runtime based

on the thresholds defined by `stage3_prefetch_bucket_size` and `reduce_bucket_size`. Offloading behavior was controlled by settings such as `stage3_max_live_parameter`, `stage3_param_persistence_threshold`, and `stage3_max_reuse_distance`, which we fine-tuned to achieve optimal performance.

In the case of Colossal-AI, we leveraged version 0.3.3 along with the Gemini Plugin to facilitate chunk-based memory management to group the parameters. This setup featured a static placement policy and also enabled the offloading of parameters and optimizer states to make large models trainable.

For Fully Sharded Data Parallel (FSDP) which is integrated within PyTorch-2.0.1, we employed the `transformer_auto_wrap_policy` to ensure that each transformer block was encapsulated within a single `FlatParameter`. We also enable CPU offloading to accommodate the training of larger models.

Gradient checkpointing is enabled for all baselines, with full checkpointing applied to every transformer block. We also compared ProTrain with FSDP using selective gradient checkpointing, as shown in Appendix D.6.

# D    FULL EXPERIMENT RESULTS

## D.1    THROUGHPUT SCALABILITY ON A100 GPUS

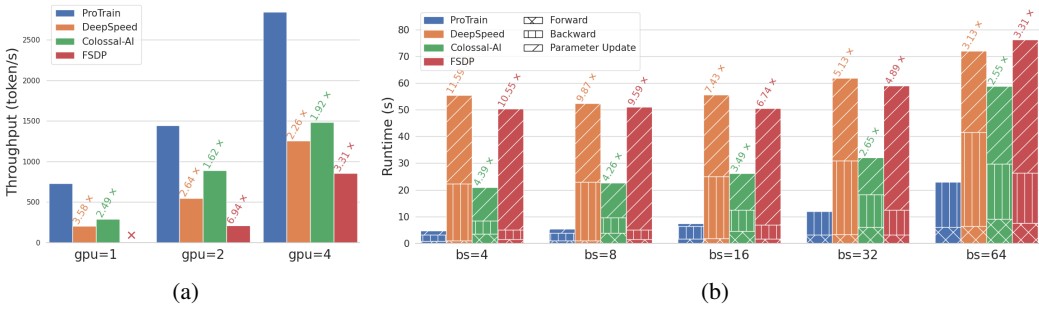

Figure 6: Scalability of performance on A100 GPUs (a) Maximum throughput across different numbers of GPUs (b) Step time breakdown for different batch sizes

Figure 6(a) presents the scalability performance of ProTrain for LLaMA 34B on four A100 GPUs compared to other frameworks. ProTrain demonstrates superior scalability, achieving a $2.49\times$ to $3.58\times$ speedup over a single GPU setup. The increased performance on A100 GPUs, compared to RTX 3090 GPUs, can be attributed to ProTrain's advanced memory management, which maximizes the utilization of the A100's larger memory capacity and higher bandwidth. This allows ProTrain to effectively scale with larger batch sizes, fully leveraging the additional resources to improve the training throughput.

Figure 6(b) breaks down the runtime per iteration into forward, backward, and parameter update phases across various batch sizes on A100 GPUs. ProTrain consistently outperforms other frameworks due to its efficient memory management and overlapping strategies. One of the most significant improvements comes from its ability to overlap CPU parameter updates with backward computations, effectively hiding the update time and reducing it to nearly zero. This optimization ensures that parameter updates do not become a bottleneck, where other other frameworks experience significant slowdowns. For instance, FSDP spends considerable time in the parameter update phase due to its use of the default Adam optimizer, which is less efficient than the optimized variants used by ProTrain. On the other hand, ProTrain significantly reduces backward execution time compared to DeepSpeed, which relies on multiple thresholds for parameter prefetching and eviction, similar to a sliding window. In DeepSpeed's approach, parameters can only be evicted after full usage, and new ones are prefetched only if they fit into the freed memory, leading to inefficient bandwidth utilization.

Overall, ProTrain delivers an average speedup of $3.47\times$ to $7.43\times$ compared to other frameworks, showcasing its superior performance across various setups.

## D.2 TRAINING THROUGHPUT W/ AND W/O OFFLOADING

Table 3: Maximum Training Throughput on four A100 GPUs w/ and w/o Offloading (Unit: token/s)

| Model | | Mistral 7B | GPT-2 10B | LLaMA 13B | GPT-2 20B |
|---|---|---|---|---|---|
| ProTrain | automatic | 11060.92 | 8266.40 | 6471.32 | 5043.75 |
| DeepSpeed | w/ | 7708.30 (1.43$\times$) | 6447.70 (1.28$\times$) | 4446.43 (1.46$\times$) | 3420.90 (1.47$\times$) |
| | w/o | 9748.03 (1.13$\times$) | 7320.50 (1.13$\times$) | 5234.92 (1.24$\times$) | OOM |
| Colossal-AI | w/ | 7279.76 (1.52$\times$) | 6848.47 (1.21$\times$) | 4980.91 (1.30$\times$) | 3892.95 (1.30$\times$) |
| | w/o | 8447.30 (1.31$\times$) | 7855.46 (1.05$\times$) | 4404.30 (1.47$\times$) | 2084.74 (2.42$\times$) |
| FSDP | w/ | 5315.81 (2.08$\times$) | 4666.03 (1.77$\times$) | 3715.12 (1.74$\times$) | 2136.16 (2.36$\times$) |
| | w/o | OOM | OOM | OOM | OOM |

Although ProTrain is designed for scenarios where the model cannot fully fit into GPU memory (requiring offloading), it also delivers excellent performance compared to baselines in non-offloading scenarios. As shown in Table 3, when DeepSpeed and Colossal-AI operate without offloading, their training throughput improves for smaller models. However, as model size increases, GPU memory becomes a bottleneck, reducing the batch size that can be trained without offloading and diminishing the performance advantage. For instance, Colossal-AI's performance on LLaMA 13B is 15% slower without offloading compared to with offloading. ProTrain addresses this bottleneck by efficiently coordinating CPU offloading and gradient checkpointing, allowing it to handle larger batch sizes and deliver better throughput. Importantly, ProTrain consistently outperforms baselines both with and without offloading, showing its versatility and adaptability across different training scenarios.

## D.3 TRAINING PERFORMANCE ON AMD MI300X GPUs

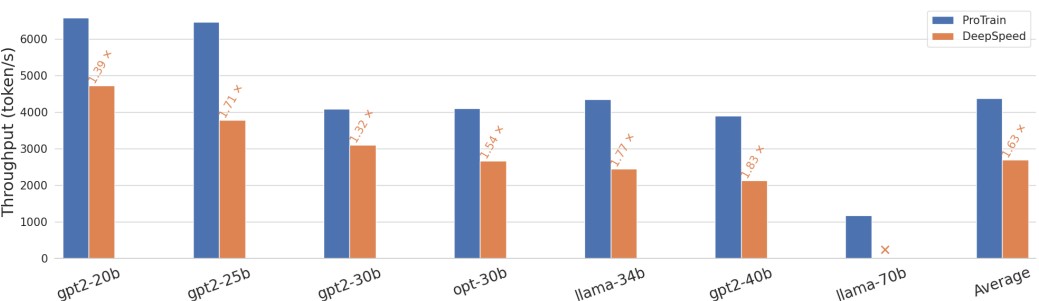

Figure 7: Maximum Training Throughput on four AMD MI300X GPUs

Figure 7 presents the throughput comparison between ProTrain and DeepSpeed across various model sizes on AMD Instinct™ MI300X GPUs, which feature 192 GB of HBM3 memory and provide 5.3 TB/s peak memory bandwidth. This extensive memory capacity and bandwidth, along with Infinity Fabric interconnect technology, enables superior multi-GPU scaling compared to RTX 3090 and A100 GPUs, making it especially advantageous for training larger models. As demonstrated in the results, ProTrain consistently surpasses DeepSpeed, with speedups ranging from $1.39\times$ to $1.83\times$ across all model configurations. This performance improvement highlights ProTrain's ability to leverage the high memory bandwidth and capacity, resulting in better hardware utilization and overall performance.

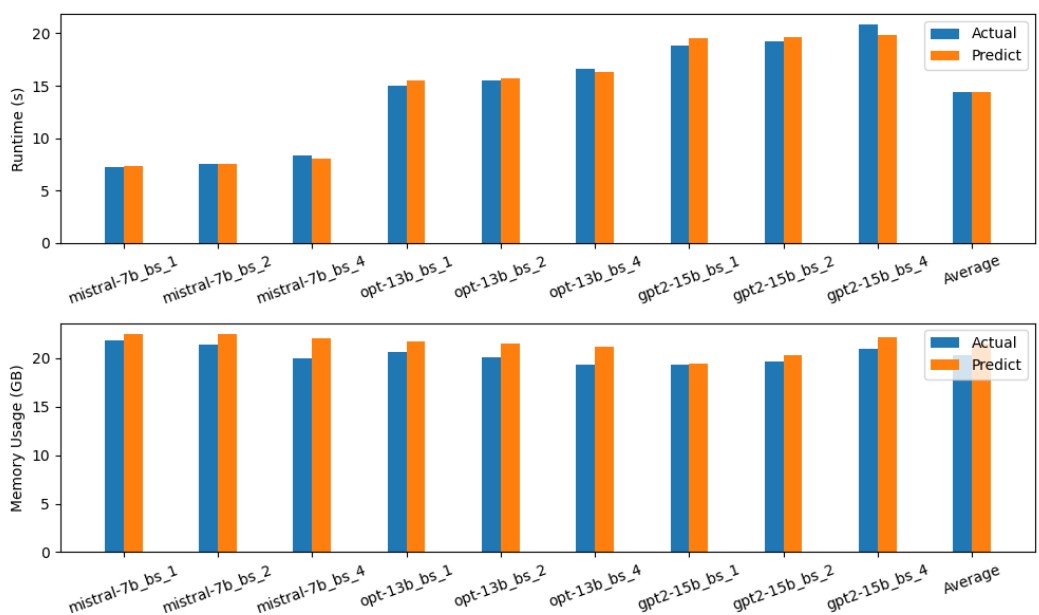

Figure 8: Comparison of Predicted vs. Actual Runtime and Peak Memory Usage for Various Models

### D.4 Effect of Runtime/Peak Memory Usage Estimator

Figure 8 compares predicted versus actual runtime and peak memory usage using ProTrain's chosen configuration on four RTX 3090 GPUs. The top chart shows the runtime prediction error does not exceed 5%, reflecting the high accuracy of the runtime estimator across different models and batch sizes. The bottom chart compares the predicted and actual peak memory usage, measured using `max_memory_allocated`. Prediction error increases slightly with larger batch sizes, typically overestimating by no more than 10%. This conservative estimation helps mitigate the risk of out-of-memory errors by accounting for memory fragmentation, thus ensuring reliable performance in diverse training conditions. Overall, these results validate ProTrain's estimators for both runtime and memory, confirming their reliability in automatic memory management.

### D.5 Search Overhead and Configuration

#### D.5.1 Search Overhead

The optimal configuration search in ProTrain is highly efficient, requiring only **0.06 seconds on average**. The profiling duration scales with the model's execution time; for example, profiling Mistral-7B with a batch size of 4 takes **3.09 seconds**, while profiling GPT-20B with the same batch size takes **5.38 seconds**. These results, obtained on RTX 3090 GPUs, highlight the minimal overhead of ProTrain's search process, enabling the effective identification of optimal configurations.

#### D.5.2 Searched Configurations

Table 4: Automatically searched configurations with the best performance.

| ID | Model, BS, HW | Chkpt / Total Blocks | Swap Blocks | Persistent / Total Chunks | Chunk Buffers |
|----|---------------|----------------------|-------------|---------------------------|---------------|
| A | GPT-1B, 8, RTX 3090s | 0 / 32 | 0 | 12 / 12 | 0 |
| B | GPT-1B, 64, RTX 3090s | 24 / 32 | 2 | 2 / 12 | 3 |
| C | GPT-1B, 64, A100s | 0 / 32 | 0 | 12 / 12 | 0 |
| D | GPT-10B, 8, RTX 3090s | 48 / 48 | 0 | 3 / 49 | 46 |
| E | GPT-10B, 8, A100s | 0 / 48 | 0 | 15 / 49 | 3 |

Table 4 summarizes the configurations automatically determined by ProTrain, showing the impact of batch size, hardware type, and model size on optimal memory management plans.

**Batch Size Impact**    When increasing the batch size from 8 (row A) to 64 (row B) on RTX 3090 GPUs, the optimal configuration changes as follows: the number of swapping blocks increases from 0 to 2, the number of gradient checkpointing blocks increases from 0 to 24, the number of persistent chunks decreases from 12 to 2, and the number of chunk buffers increases from 0 to 3. These configurations align with runtime execution patterns. A larger batch size increases the computation intensity of the forward and backward pass, making it possible for parameter uploads to be fully hidden by the computation. As a result, as the batch size increases, ProTrain prioritizes offloading and thus uses fewer persistent chunks and more swapping blocks to save GPU memory. ProTrain selects 2 swapping blocks as the swapping overhead for 2 blocks can be effectively overlapped with computation without impacting parameter prefetching.

**Hardware Impact**    When training GPT-1B with BS=8 (row A), GPU memory is sufficient on both A100 and RTX 3090 hardware. Therefore, no offloading or activation checkpointing is required, and the configurations are identical. For GPT-1B with BS=64 (rows B and C), A100 GPUs have sufficient memory, while RTX 3090 requires offloading and checkpointing, leading to different configuration choices. For GPT-10B with BS=8 (rows D and E), both hardware lack sufficient memory, but their configurations differ due to varying runtime patterns. RTX 3090s, lacking NVLink and being communication-bound for NCCL operations, use checkpointing for all blocks to allocate more space for larger chunk buffers and persistent chunks, reducing parameter gathering overhead that cannot be fully hidden by computation. In contrast, A100 GPUs, equipped with NVLink and thus have a much higher communication bandwidth, retain all activations and save memory by offloading model states, using fewer chunk buffers and persistent chunks.

**Model Size Impact**    The table shows that different model sizes require different configuration combinations. As the model size increases, there is generally more offloading (fewer persistent chunks and chunk buffers, more swapping blocks) and more gradient checkpointing (more checkpointing blocks). These adjustments optimize memory, enabling efficient training and fine-tuning of larger models within hardware limits.

## D.6    COMPARISON OF PROTRAIN AND FSDP WITH SELECTIVE CHECKPOINTING

Table 5: Maximum Training Throughput of FSDP with and without selective checkpointing and ProTrain (Unit: tokens/s)

| Model | FSDP + Selective Checkpointing | FSDP - Selective Checkpointing | ProTrain |
|---|---|---|---|
| LLaMA-13B | 3996.67 (1.00×) | 3715.12 (0.93×) | 6471.32 (1.62×) |
| GPT-20B | 2392.17 (1.00×) | 2136.16 (0.89×) | 5043.75 (2.11×) |
| GPT-30B | 1383.52 (1.00×) | 1307.88 (0.95×) | 3431.38 (2.48×) |
| OPT-30B | 1621.85 (1.00×) | 1342.40 (0.83×) | 3266.02 (2.01×) |
| LLaMA-34B | 1247.25 (1.00×) | 1024.23 (0.82×) | 2845.18 (2.28×) |
| GPT-40B | 1143.06 (1.00×) | 1208.68 (1.06×) | 2723.50 (2.38×) |

The FSDP baseline initially applied gradient checkpointing to all blocks. To assess the potential benefits of selective gradient checkpointing, we re-evaluated FSDP with this approach on both RTX 3090 and A100 GPUs. On RTX 3090 GPUs, selective checkpointing does not improve throughput because execution is communication-bound, making recomputation savings ineffective. In contrast, on A100 GPUs, selective checkpointing improves throughput for all models except GPT-40B, which fails to scale due to GPU OOM issues. Table 5 shows the maximum throughput on A100 GPUs for three configurations: (A) FSDP with Selective Checkpointing, (B) FSDP without Selective Checkpointing, and (C) ProTrain. Although FSDP with selective checkpointing improves performance compared to the configuration without it, ProTrain still outperforms it by effectively balancing offloading and checkpointing, enabling better utilization of hardware resources and higher throughput.

# E    RELATED WORK

**Swapping and Recomputation**    Swapping Rhu et al. (2016); Le et al. (2018); Huang et al. (2020); Ren et al. (2021); Sun et al. (2022) is a commonly employed technique which leverages external memory such as CPU memory to offload tensors, thereby expanding the available memory for training. Traditional swapping methods mainly focus on offloading activations, SwapAdvisor Huang et al. (2020) extends it to parameters and ZeRO-offload Ren et al. (2021) further extends it to optimizer states. Recomputation Chen et al. (2016); Jain et al. (2020); Herrmann et al. (2019); Zhao et al. (2023a); Korthikanti et al. (2023), also known as gradient checkpointing, is another widely used technique that trades additional recompute time during backward pass for reduced memory usage of activations. Initially, Chen et al. Chen et al. (2016) focuses on homogeneous sequential networks, and subsequent studies Jain et al. (2020); Herrmann et al. (2019) extended its applicability to heterogeneous networks. Considering the scale and complexity of Transformers, which often contain numerous layers, previous approaches become less efficient. Therefore, Rockmate Zhao et al. (2023a) optimizes the plan generation by partitioning models into fine-grained blocks. NVIDIA further proposes selective activation recomputation which checkpoints and recomputes parts of layers Korthikanti et al. (2023). To get the best of both worlds, some works Peng et al. (2020); Beaumont et al. (2021); Nie et al. (2022) jointly optimize swapping and recomputation, whereas ProTrain differentiates itself by tailoring to fit the specific structure of transformers.

**ZeRO Techniques.**    ProTrain adopts ZeRO to manage model states. The Zero Redundancy Optimizer (ZeRO) Rajbhandari et al. (2020) distributes model states across multiple GPUs to reduce memory pressure of each GPU. ZeRO operates in three stages: ZeRO-1 partitions optimizer states across GPUs; ZeRO-2 extends this by also distributing gradients; and ZeRO-3 further divides the parameters, which are required to be gathered before forward/backward computation. The ZeRO techniques have been integrated into state-of-the-art frameworks such as DeepSpeed Rasley et al. (2020), FSDP Zhao et al. (2023b), and Colossal-AI Li et al. (2023), each differing in their parameter organization to optimize bandwidth utilization. Unlike DeepSpeed and FSDP, which require manual configuration for parameter grouping, Colossal-AI automatically groups parameters into chunks and dynamically adjusts their size according to the model's scale. This chunk-based method, inspired by PatrickStar Fang et al. (2022), is also adopted in ProTrain.

**GPU Memory Management**    Deep learning frameworks, such as PyTorch Paszke et al. (2019) and TensorFlow Abadi et al. (2016), utilize caching allocators for efficient memory management. However, these frameworks often face memory fragmentation issues, particularly when integrating memory-saving techniques like swapping, recomputation, and parallelization, which hurts allocation efficiency. To address this, two main approaches have been proposed. The first is profiling-guided optimization Sekiyama et al. (2018); Steiner et al. (2022; 2023), which leverages the repetitive and predictable nature of memory allocation patterns during training. This method traces and analyzes tensor allocations and deallocations to optimize tensor placement, thus improving memory efficiency. Alternatively, GMLake Guo et al. (2024) introduces Virtual Memory Stitching, a technique that merges non-contiguous memory blocks, thereby reducing memory fragmentation at the operating system level. These approaches are orthogonal to ProTrain's method. Angel-PTM Nie et al. (2023) adopts a page-based memory management strategy that partitions model states to reduce the memory fragmentation. In contrast, ProTrain designs a new chunk-based memory management inspired by PatrickStar Fang et al. (2022) grouping model states into chunks that align with the runtime execution order, which not only improves bandwidth utilization but also enhances memory locality.

**Overlapping Computation and Communication**    There are numerous work on overlapping computation and communication, with many studies Mahajan et al. (2023); Hashemi et al. (2019); Peng et al. (2019); Jangda et al. (2022); Chen et al. (2024) focus on substituting, splitting, and scheduling complex operators to achieve fine-grained overlapping. CoCoNet Jangda et al. (2022) enhances lower-level operator optimization, while Centauri Chen et al. (2024) extends this to graph-level scheduling, offering a more hierarchical abstraction. Despite these advances, most research focuses on the optimization of collective communication operations in distributed cases. However, ProTrain also considers the communication between CPU and GPU under limited GPU memory conditions, making it orthogonal to existing research.

**Training Frameworks for Transformers** In response to the growing demand for efficient training of transformers, several specialized frameworks have been developed, each offering unique features and optimizations. DeepSpeed Rasley et al. (2020) by Microsoft enhances training efficiency through ZeRO series techniques Rajbhandari et al. (2020); Ren et al. (2021); Rajbhandari et al. (2021) and supports various parallelism strategies, swapping, and recomputation. Colossal-AI Li et al. (2023) from HPC-AI Tech, which offering similar features, distinguishes itself with a chunk-based memory management approach Fang et al. (2022), which our work adopts. Megatron-LM Shoeybi et al. (2019) by NVIDIA, on the other hand, specializes in model parallelism. These frameworks are designed for large-scale transformer training, complemented by academic efforts Sun et al. (2022); Li et al. (2022); Feng et al. (2023) to facilitate training on smaller systems.

