# OpenReview forum: "ProTrain: Efficient LLM Training via Automatic Memory Management"
_ICLR.cc/2025/Conference — Submitted to ICLR 2025_

### Official Review · Reviewer_EEcS · 2024-10-28

**Soundness:** 2
**Presentation:** 2
**Contribution:** 2
**Rating:** 5
**Confidence:** 2

**Summary:**

Training large language models (LLMs) demands intricate memory management strategies. However, current memory management approaches often rely on manually tuned configurations, which results in suboptimal hardware utilization and performance. In this paper, the authors introduce ProTrain, an advanced training system that autonomously develops memory management policies to efficiently support the training of various LLMs across diverse hardware platforms. ProTrain reframes memory management as an optimization problem with a limited set of tunable parameters that are optimized using a cost model. Additionally, it incorporates a runtime profiler to provide precise estimations of latency, memory consumption, and I/O bandwidth, thus enhancing the accuracy of the cost model.

**Strengths:**

1. The paper works on an important topic.
2. The techniques are evaluated on various LLMs.
3. The techniques are tested on different hardware platforms.

**Weaknesses:**

1. The techniques proposed by this paper cannot be applied on the real-world LLM training systems, since it includes off-loading. At most, these techniques can be used for only fine-tuning of the LLMs.

2. The paper lacks of motivaiton. Although this paper claims that "existing frameworks often depend on manual tuning of memory management settings, leading to inefficient hardware utilization and suboptimal performance". The paper did not show how bad existing frameworks are.

3. Why not LORA? Maybe LORA can achieve the same performance but consumes much less training overhead.

4. The paper is not well-written. Many key concepts are not well-introduced. For example, off-loading, and activation recomputation.

**Questions:**

1. What is the design overhead of ProTrain?

2. Why is managing activation swapping and gradient checkpointing operations at the transformer block level a better policy? Is it possible to achieve better thourghput with a more fine-grained management scheme, e.g., the smallest granularity of managing activation swapping and gradient checkpointing operations is 1/3 of the transformer block?

---

> ### Author Response · Authors · 2024-11-18
>
> Q1: The paper lacks motivation. Although this paper claims that "existing frameworks often depend on manual tuning of memory management settings, leading to inefficient hardware utilization and suboptimal performance". The paper did not show how bad existing frameworks are.
>
> A1: Thank you for highlighting this concern. Take DeepSpeed as an example, it requires users to manually specify at least five configurations for offloading and prefetching, including `stage3_max_live_parameters`, `stage3_max_reuse_distance`, `stage3_prefetch_bucket_size`, `stage3_param_persistence_threshold`, and `reduce_bucket_size`. These configurations are all numerical values that are extremely hard to tune compared to boolean settings. For instance, running GPT-10B on RTX 3090 with the default configuration utilizes only 35.6% of the GPU memory and results in a 1.18x performance slowdown compared to the optimized configuration. Additionally, optimal configurations for A100s cannot be directly applied to RTX 3090s due to a higher risk of GPU OOM. Finding the right settings often requires an exhaustive grid search, which can take hours or even days.
>
> Similar to DeepSpeed, ColossalAI requires users to tune numerous configurations. FSDP, on the other hand, simplifies configuration settings but offers less flexibility for offloading, supporting only two options: offload all parameters or none. Motivated by the need for both flexibility and simplicity, ProTrain automates memory management, eliminating the need for manual tuning while optimizing memory usage and runtime performance.
>
> Q2: The techniques proposed by this paper cannot be applied on the real-world LLM training systems, since it includes off-loading. At most, these techniques can be used for only fine-tuning of the LLMs.
>
> A2: Thank you for your feedback. ProTrain specifically targets scenarios where GPU memory becomes a bottleneck and cannot accommodate all the data required for model training. Our evaluation demonstrates that ProTrain achieves better throughput compared to existing systems in such cases.
>
> Furthermore, offloading can still be beneficial for real-world LLM training systems. As shown in Appendix D.2, we compared ProTrain against baselines with and without offloading. For smaller models, baselines without offloading achieve higher throughput than those with offloading. However, as model size increases and memory becomes a bottleneck, offloading becomes advantageous. ProTrain outperforms baselines both with and without offloading because it effectively coordinates CPU offloading and activation checkpointing, allowing it to handle larger batch sizes and achieve higher throughput.
>
> Q3: Why not LORA? Maybe LORA can achieve the same performance but consumes much less training overhead.
>
> A3: Thank you for the suggestion. LoRA is an effective approach for reducing training overhead, particularly for parameter-efficient fine-tuning. However, as shown in Table 2 of [1], LoRA may sacrifice accuracy compared to full fine-tuning on certain datasets, such as MNLI, CoLA, and QQP. ProTrain offers an alternative solution for users who require full fine-tuning.
>
> Moreover, LoRA optimizes memory usage by altering the training algorithm, while ProTrain focuses on system-level memory management. The two approaches are orthogonal and can work together to further improve training efficiency.
>
> [1] Edward J Hu, Yelong Shen, Phillip Wallis, Zeyuan Allen-Zhu, Yuanzhi Li, Shean Wang, Lu Wang, and Weizhu Chen. LoRA: Low-rank adaptation of large language models. In International Conference on Learning Representations, 2022
>
> Q4: The paper is not well-written. Many key concepts are not well-introduced. For example, off-loading, and activation recomputation.
>
> A4: Thank you for the comment. Offloading refers to transferring data from GPU to CPU memory to handle memory constraints, while activation recomputation recalculates intermediate activations during the backward pass to save memory. We will revise the paper to introduce these concepts more clearly for better understanding.
>
> Q5: What is the design overhead of ProTrain?
>
> A5: If "design overhead" refers to the search overhead for finding the memory management plan, please see Common Question-2 for details.

---

> > ### Author Response · Authors · 2024-11-18
> >
> > Q6: Why is managing activation swapping and gradient checkpointing operations at the transformer block level a better policy? Is it possible to achieve better throughput with a more fine-grained management scheme, e.g., the smallest granularity of managing activation swapping and gradient checkpointing operations is 1/3 of the transformer block?
> >
> > A6: As an LLM typically consists of identical transformer blocks, ProTrain’s design space uses a transformer block as a basic unit for swapping or checkpointing. The block-level implementation has two benefits: (1) Users need to insert checkpointing and swapping instructions in the code to support these functions. The block-level implementation keeps user code changes minimal compared to a finer granularity. (2) Block-level implementation is easier to implement for framework developers. A finer granularity (e.g., 1/3 of the transformer blocks or tensor-level) increases implementation complexity and scalability challenges, making it difficult to adopt in practice. To the best of our knowledge, we haven’t seen tensor-level management adopted in well-established LLM training frameworks such as DeepSpeed, PyTorch, ColossalAI, and Megatron so far.

---

> > ### Comment · Reviewer_EEcS · 2024-12-02
> >
> > Thank you for your response. I still think this paper needs some improvement. I will keep my current rating.

---

### Official Review · Reviewer_9uNi · 2024-11-01

**Soundness:** 3
**Presentation:** 2
**Contribution:** 3
**Rating:** 6
**Confidence:** 3

**Summary:**

This paper introduces **ProTrain**, a training framework aimed at optimizing large language model (LLM) training on memory-limited hardware through **automated** data management across GPUs and CPUs, using a few tunable parameters. ProTrain’s design is based on two key insights: (a) **Model states**—prioritizing GPU residence for the initial layers minimizes latency due to cold starts and backward computation order; (b) **Activations**—retaining the final layers on GPU facilitates immediate access during backward passes. For all other model states and activations, ProTrain employs **offloading (swapping)**  and **checkpointing** to reduce memory usage. ProTrain’s memory and computation profiling enables efficient overlap of memory operations with computation, maximizing hardware utilization. Evaluations show that ProTrain supports models up to **1.64x larger** and achieves a **1.43x throughput improvement** over state-of-the-art solutions.

**Strengths:**

1. The dual-chunk design for model states and interleaved organization for activation management effectively reduce memory overhead by transforming memory management into a configuration task with a few tunable parameters.
2. The authors implement a memory and computation profiling system that supports automated memory management during training, optimizing resource allocation without manual intervention.
3. ProTrain demonstrates promising results, not only in expanding the trainable model size but also in boosting throughput. Integrating this system into existing platforms could significantly enhance training efficiency.

**Weaknesses:**

1. The paper primarily addresses training with data parallelism; however, expanding ProTrain to pipeline parallelism would improve its applicability, especially as modern models rarely fit on a single GPU during training.
2. Forward pass efficiency also depends on batch size, and a limited parallel scale may not sufficiently overlap memory transfer with computation. A more detailed micro-benchmark could clarify how effectively ProTrain overlaps memory transfer with computation.

**Questions:**

1. **Persistent Chunks?** Given that the model states for the initial chunk are only required at the start and end of the training pipeline, what is the advantage of keeping them persistently on the GPU? In two consecutive iterations, it’s understandable that updating these states on the GPU is efficient since they’re reused right after the first iteration. However, why not evict these states during the forward pass to free up space for other data, swapping them back only at the end of the backward pass to optimize memory usage, just like the other chunks?
2. **Difference from LRU?** Although the dual observations on managing model states and activations are valuable, it’s unclear how this approach offers distinct benefits over a computation-aware Least Recently Used (LRU) caching strategy, which also prioritizes recent data retention on the GPU within the stacked training pipeline. Could you clarify this difference?
3. **Portability Concerns** Since ProTrain is built independently on top of PyTorch, how can developers who are already using frameworks like DeepSpeed or Colossal-AI integrate ProTrain into their existing codebases?

---

> ### Author Response · Authors · 2024-11-18
>
> Q1: The paper primarily addresses training with data parallelism; however, expanding ProTrain to pipeline parallelism would improve its applicability, especially as modern models rarely fit on a single GPU during training.
>
> A1: Yes, ProTrain currently focuses on data parallelism, and we agree that supporting pipeline parallelism would enhance its applicability. When extending ProTrain to pipeline parallelism, most existing functions can be reused. The primary modification lies in the communication module: gradient synchronization (used in data parallelism) needs to be changed to activation passing between pipeline stages.
>
> Q2: Forward pass efficiency also depends on batch size, and a limited parallel scale may not sufficiently overlap memory transfer with computation. A more detailed micro-benchmark could clarify how effectively ProTrain overlaps memory transfer with computation.
>
> A2: Thank you for this great point! Similar to other LLM training frameworks, ProTrain aims to improve training efficiency for a user-specified batch size and model configuration. While communication can sometimes become the bottleneck and may not be fully overlapped with computation, ProTrain maximizes the overlap to ensure efficiency. Additionally, our runtime cost model considers scenarios where communication overhead exceeds computation, enabling accurate runtime estimates. Within this context, it is insightful to analyze the searched configurations ProTrain identifies as batch size varies, as discussed in Common Question-2.
>
> Q3: Persistent Chunks? Given that the model states for the initial chunk are only required at the start and end of the training pipeline, what is the advantage of keeping them persistently on the GPU? In two consecutive iterations, it’s understandable that updating these states on the GPU is efficient since they’re reused right after the first iteration. However, why not evict these states during the forward pass to free up space for other data, swapping them back only at the end of the backward pass to optimize memory usage, just like the other chunks?
>
> A3: The design of using initial chunks as persistent chunks is based on the following rationale:
>
> 1. **Backward computation timing**: Initial chunks perform backward computation at the end, offering limited opportunities to overlap parameter updates on the CPU. Therefore, keeping their model states on the GPU and updating them directly reduces delays and improves efficiency.
>
> 2. **Bandwidth limitations**: Model states consume significant memory, but bandwidth is limited. Minimizing unnecessary back-and-forth data transfers reduces communication overhead and saves bandwidth.
>
> 3. **Profiling-driven decisions**: The number of persistent chunks is determined using profiling insights and cost models. When GPU memory is extremely constrained, as identified during profiling, ProTrain’s cost model will determine the optimal configuration using zero persistent chunks.
>
> Q4: Difference from LRU? Although the dual observations on managing model states and activations are valuable, it’s unclear how this approach offers distinct benefits over a computation-aware Least Recently Used (LRU) caching strategy, which also prioritizes recent data retention on the GPU within the stacked training pipeline. Could you clarify this difference?
>
> A4: ProTrain incorporates the LRU concept for managing both model states and activations. However, it manages these two components separately because of the difference in the optimization techniques available to model states and activations: ZeRO and offloading are applicable to model states, while activation checkpointing and offloading are applied to activations.
>
> Q5: Portability Concerns Since ProTrain is built independently on top of PyTorch, how can developers who are already using frameworks like DeepSpeed or Colossal-AI integrate ProTrain into their existing codebases?
>
> A5: Thank you for raising this concern. Although ProTrain is a new training framework itself, the techniques implemented in ProTrain can be easily integrated into existing frameworks. Specifically, ProTrain employs a chunk-based management approach for model states, a block-wise management strategy for activations, and a model-wise runtime profiler. The block-wise management approach is straightforward to integrate into existing frameworks, as transformer architectures typically follow a similar structure.
>
> For chunk-based management, integration is relatively simple with frameworks like Colossal-AI and PyTorch FSDP, as they group model states into chunks during initialization. DeepSpeed, however, dynamically groups model states at runtime based on configurable thresholds, making alignment with ProTrain’s chunk-based design more challenging and requiring additional adjustments.
>
> The model-wise runtime profiler is a separate component that can be used independently for collecting memory and runtime usage.

---

> > ### Comment · Reviewer_9uNi · 2024-11-30
> >
> > Thanks for your clarification. I would keep my rating.

---

### Official Review · Reviewer_sdYt · 2024-11-03

**Soundness:** 2
**Presentation:** 3
**Contribution:** 2
**Rating:** 5
**Confidence:** 3

**Summary:**

This paper proposed a cost model to evaluate different memory technics to achive best perf under memory constraint. The memory technics include 1) parameters offloading, 2) parameters all-gather and gradients reduce-scatter, 3) activation checkpoint, 4) activation offloading. It also proposed a dual-chunk heuristics that persist first layer on gpu to avoid exposed all-gathers.

The author also built 2 fundemental components to support cost model: 1) runtime estimator for forward, backward, gpu/cpu optimizer, 2) peak memory estimator

**Strengths:**

1. The search space covers wide spectrum of memory technics including ZeRO, param offloading, activation offloading, activation checkpoint, gpu persistent parameters, and optimizer in the backward

2. Runtime estimator and peak memory estimator are foudemental work that can be useful in general, including OOM prevention and offline simulation

**Weaknesses:**

1. The explanation for maximun trainable model size seem to be vague. "Some frameworks fail to scale model sizes with more GPUs,
primarily due to inefficiencies in handling model initialization across devices.". It would be great to be concrete about this. How exactly each baseline is configured and why they fail. Is it because of meta/cpu/gpu init, or cpu offloading

2. For training throughput, it would be great to cover the configuration for baseline - whether they use cpu offloading, what are the activation checkpoints, how parameters are grouped for all-gather and reduce-scatter

**Questions:**

1. for optimizer in the backward, curious how does it compose with gradient norm clipping? for LLM training, it seems quite critical to clip gradients base on total norms to avoid numeric issues. If I understand it correctly, we perform optimizer step for each gradient or layer now without knowing the total gradient norms

2. for "Some frameworks fail to scale model sizes with more GPUs, primarily due to inefficiencies in handling model initialization across devices.", curious what are the foundmental issue for FSDP to only support 1B model at most? Is it init model on gpu before applying fsdp?

3. it would be great to summarize the exact combination of different tricks when they reach the best memory and perf.

---

> ### Author Response · Authors · 2024-11-18
>
> Q1: The explanation for maximum trainable model size seem to be vague. "Some frameworks fail to scale model sizes with more GPUs, primarily due to inefficiencies in handling model initialization across devices.". It would be great to be concrete about this. How exactly each baseline is configured and why they fail. Is it because of meta/cpu/gpu init, or cpu offloading? curious what are the fundamental issue for FSDP to only support 1B model at most? Is it init model on gpu before applying fsdp?
>
> A1: Thank you for pointing this out. All baselines perform initialization in CPU memory. DeepSpeed and ColossalAI fail when the parameter size exceeds CPU memory capacity. FSDP fails when wrapping the model with `FullyShardedDataParallel` (a required step for using FSDP to shard model parameters across devices) because it initializes each parameter group as a  `FlatParameter` on the GPU before offloading it to the CPU. Due to improper handling of parameter references, CUDA memory is not released until initialization is completed.
> In contrast, ProTrain utilizes both CPU and GPU memory during initialization, increasing the total initialization capacity.
>
> Q2: For training throughput, it would be great to cover the configuration for baseline - whether they use cpu offloading, what are the activation checkpoints, how parameters are grouped for all-gather and reduce-scatter
>
> A2: Thank you for the suggestion. In Appendix C.3, we provide descriptions of the baseline configurations, which are applied to all experiments unless otherwise specified. Specifically:
>
> 1. **CPU Offloading**: As mentioned in A1, all baselines utilize CPU offloading. ColossalAI and DeepSpeed allow configurable offloading ratios, while FSDP supports only two options: either no CPU offloading or complete CPU offloading.
>
> 2. **Activation Checkpointing**: Activation checkpointing is enabled for all baselines, with full checkpointing applied to all transformer blocks.
>
> 3. **Parameter Grouping**: ColossalAI and FSDP group parameters during initialization, using these groups for all-gather and reduce-scatter. ColossalAI searches for an optimal group size to minimize memory fragmentation, while FSDP combines parameters within a transformer block into a single `FlatParameter`. In contrast, DeepSpeed groups parameters and gradients at runtime based on user-specified thresholds (`stage3_prefetch_bucket_size` for all-gather and `reduce_bucket_size` for reduce-scatter).
>
> Q3: for optimizer in the backward, curious how does it compose with gradient norm clipping? for LLM training, it seems quite critical to clip gradients base on total norms to avoid numeric issues. If I understand it correctly, we perform optimizer step for each gradient or layer now without knowing the total gradient norms
>
> A3: Thank you for the question. Yes, gradient clipping based on total norms is critical to avoiding numeric issues. However, since ProTrain executes part of the optimizer step earlier, it cannot compute total gradient norms directly. To address this, ProTrain supports alternative methods such as value-based clipping and component-wise gradient norm clipping, which have been shown to improve fine-tuning performance, as demonstrated in Table 1 of [1].
>
> [1] Chenghao Yang and Xuezhe Ma. Improving stability of fine-tuning pretrained language models via component-wise gradient norm clipping. In Proceedings of the 2022 Conference on Empirical Methods in Natural Language Processing, 2022.
>
> Q4: it would be great to summarize the exact combination of different tricks when they reach the best memory and perf.
>
> A4: Thank you for your suggestion! Please refer to Common Question-2.

---

> > ### Comment · Reviewer_sdYt · 2024-11-27
> >
> > thanks for addressing all of my questions. A1 is espectially useful for me to understand the nuances among DeepSpeed, ColossalAI and FSDP. I am keeping my rating as it is

---

### Official Review · Reviewer_hSX9 · 2024-11-03

**Soundness:** 2
**Presentation:** 3
**Contribution:** 2
**Rating:** 5
**Confidence:** 5

**Summary:**

This paper introduced ProTrain, a novel training system designed to simplify the training process through automatic memory management. ProTrain highlights the significance of precise memory usage and runtime data gathered through memory-aware, model-wise profiling to build high-fidelity cost models, along with the careful abstraction of configuration parameters from memory management strategies to automate optimal configuration search.The basic idea of ProTrain is to abstract memory management strategies into a few tunable configuration parameters. ProTrain then builds runtime and memory usage estimators that quantify the impacts of these configuration parameters on training performance. These cost models, informed with accurate profiling information on latency, memory, and I/O bandwidth, allow ProTrain to search for the optimal memory management strategy that minimizes runtime while ensuring the peak memory consumption meets the hardware constraints.

**Strengths:**

Thank you for submitting this paper to ICLR!

1. Important problem. Reducing memory consumption in LLM training is important and challenging.

2. Clear written, well-organized, and easy to follow.

**Weaknesses:**

1. Limited technical innovation. The technical contribution of the paper is somewhat restricted, as it relies on the application of ZeRO, Offloading and Activation Checkpointing. Their combination is not novel enough, and the large search space is not well justified (can be easily pruned).

2. Small-scale experiments: The experiments are conducted on a single node server, with only 4 GPUs. Although authors have some discussion on the two V100 nodes, the experiments are not sufficient to support the LLM training scenario. The paper should provide more extensive experiments on larger scales to demonstrate the scalability of the proposed method.

3. Unfair baseline comparison: The paper compares the proposed method with the vanilla FSDP (i.e., ZeRO-3 etc). However, Pytorch already support selective activation checkpointing. The paper should compare with the Pytorch's combination of FSDP, Offloading and Activation Checkpointing.

**Questions:**

1. Please provide some results about search overhead and the searched configuration.

2. Please discuss the novelty part. I think the technical contribution of the paper is somewhat restricted, as it relies on the application of ZeRO, Offloading and Activation Checkpointing. Their combination is not novel enough. Could you please highlight the challenge of combining them?

---

> ### Author Response · Authors · 2024-11-18
>
> Q1: Please provide some results about search overhead and the searched configuration.
>
> A1: Please refer to Common Question-2.
>
> Q2: I think the technical contribution of the paper is somewhat restricted, as it relies on the application of ZeRO, Offloading and Activation Checkpointing. Their combination is not novel enough. Could you please highlight the challenge of combining them?
>
> A2: The challenge in combining ZeRO, offloading, and activation checkpointing lies in efficiently and automatically identifying whether a tensor should use offloading or activation checkpointing. The search space is huge and ProTrain prunes the search space by leveraging transformers’ architectural features and new optimization opportunities identified in this work.
>
> (1) As an LLM typically consists of identical transformer blocks, ProTrain’s design space uses a transformer block as a basic unit for swapping or checkpointing.
>
> (2) Since ZeRO and offloading apply to model states while offloading and activation checkpointing apply to activations, ProTrain separately manages model states using chunks and activations using blocks.
>
> (3) ProTrain also identifies a few new optimization opportunities when combining ZeRO, Offloading, and Activation Checkpointing in memory management and exposes them as additional tuning knobs to identify a better policy. (a) **ProTrain introduces a novel dual-chunk system for model states**. The design improves dynamic memory management by introducing persistent chunks and reusable chunk buffers. It also helps establish more accurate estimates for runtime and memory usage. (b) **ProTrain interleaves Swapping and Checkpointing**. The design is based on the rationale that, if the swapping overhead of a transformer block can be hidden by the forward and backward computation, the block would prefer swapping rather than checkpointing. Otherwise, the block should use checkpointing to reduce CPU-GPU bandwidth contention.
>
> Q3: Small-scale experiments: The experiments are conducted on a single node server, with only 4 GPUs. Although authors have some discussion on the two V100 nodes, the experiments are not sufficient to support the LLM training scenario. The paper should provide more extensive experiments on larger scales to demonstrate the scalability of the proposed method.
>
> A3: Thank you for your feedback. Our experiments were mainly conducted on a single-node server with 4 GPUs, and we agree that larger-scale tests would better demonstrate ProTrain’s scalability. Currently, the largest resource accessible to us is a 4-node setup of 80GB A100 GPUs, with 4 GPUs per node. In this configuration, ProTrain can train GPT-175B with a batch size of 256 and achieve 2218.70 token/s throughput, which is 2.09x higher than ColossalAI, and 2.58x higher than DeepSpeed.
>
>
> Q4: Unfair baseline comparison: The paper compares the proposed method with the vanilla FSDP (i.e., ZeRO-3 etc). However, Pytorch already support selective activation checkpointing. The paper should compare with the Pytorch's combination of FSDP, Offloading and Activation Checkpointing.
>
> A4: Thank you for pointing this out. In our experiments, the FSDP baseline included ZeRO-3, offloading, and activation checkpointing for all blocks. However, we did not include selective activation checkpointing. To ensure a fair comparison, we re-tested FSDP with selective checkpointing to get the maximum achievable throughput on RTX 3090 and A100 GPUs. For selective activation checkpointing, we applied checkpointing to an increasing number of interleaved blocks until GPU OOM.
>
> On RTX 3090s, selective activation checkpointing does not improve throughput because the execution is communication-bound. In this case, computation overhead cannot effectively overlap communication overhead, so saving recomputation time does not lead to performance gains.
>
> On A100 GPUs, selective activation checkpointing benefits all models except GPT-40B, which fails to scale due to GPU OOM. The table below presents the maximum achievable throughput for the following cases: (A) FSDP with Selective Checkpointing (B) FSDP without Selective Checkpointing (C) ProTrain. It also includes the speedup relative to FSDP with Selective Checkpointing. We will update these results in our paper.
>
> | Model (Unit: token/s) | (A)             | (B)             | (C)             |
> | ---------------------------- | --------------- | --------------- | --------------- |
> | LLaMA-13B                    | 3996.67 (1.00x) | 3715.12 (0.93x) | 6471.32 (1.62x) |
> | GPT-20B                      | 2392.17 (1.00x) | 2136.16 (0.89x) | 5043.75 (2.11x) |
> | GPT-30B                      | 1383.52 (1.00x) | 1307.88 (0.95x) | 3431.38 (2.48x) |
> | OPT-30B                      | 1621.85 (1.00x) | 1342.40 (0.83x) | 3266.02 (2.01x) |
> | LLaMA-34B                    | 1247.25 (1.00x) | 1024.23 (0.82x) | 2845.18 (2.28x) |
> | GPT-40B                      | 1143.06 (1.00x) | 1208.68 (1.06x) | 2723.50 (2.38x) |

---

> > ### Comment · Reviewer_hSX9 · 2024-12-01
> >
> > Thank you for your response. My concerns have been addressed. I will keep my score at the current rating.

---

### Author Response · Authors · 2024-11-18

## Common Questions

### Question-1: Novelty of ProTrain

Our novelty includes the following aspects:

1. **Automation via Cost Models**: ProTrain automatically and efficiently identifies the best memory management policy for a given LLM architecture and hardware setup. Unlike existing systems, which require extensive manual tuning and grid search (often taking hours or days) to meet memory and performance needs, ProTrain completes this search within seconds (as shown in Common Question-2), streamlining the setup process and maximizing training throughput. Such automation is enabled by **two cost models** that estimate the memory consumption and runtime depending on configurations.

2. **Novel Design Space**: ProTrain identifies a few new optimization opportunities when combining ZeRO, Offloading, and Activation Checkpointing in memory management and exposes them as additional tuning knobs to identify a better policy. (a) **ProTrain introduces a novel dual-chunk system for model states**. The design improves dynamic memory management by introducing persistent chunks and reusable chunk buffers. It also helps establish more accurate estimates on runtime and memory usage during profiling. (b) **ProTrain interleaves Swapping and Checkpointing**. The design is based on the rationale that, if the swapping overhead of a transformer block can be hidden by the forward and backward computation, the block would prefer swapping rather than checkpointing. Otherwise, the block should use checkpointing to reduce CPU-GPU bandwidth contention.

3. **Model-wise Profiling**: ProTrain designed a novel model-wise runtime profiler that has two distinctive features. First, it is the first to consider temporary tensors and unhookable operators which can contribute up to 17.2% of peak memory usage but are often ignored by existing methods. Second, it supports profiling for large models that cannot fully fit into GPU memory by separating the memory consumption from static tensors, which are known as apriori, from the peak memory consumption estimation. The runtime profiler allows ProTrain to get precise peak memory consumption estimation for cost models, as demonstrated in Figure 5(b) and Figure 8 in Appendix D.4.

### Question-2: Search overhead and searched configuration

Most of the time in obtaining the optimal memory management plan is spent on profiling, as **search itself takes only 0.06s on average**. Profiling time depends on the model's execution time; for instance, profiling Mistral-7B with batch size 4 takes 3.09s, while GPT-20B with batch size 4 takes 5.38s. These results were obtained using RTX 3090 GPUs.

The table below summarizes the automatically searched configurations with the best performance. It shows that model type, batch size, and hardware combinations can lead to varying configurations:

| ID  | Model and Hardware                  | # of checkpointing blocks / # of blocks | # of swapping blocks | # of persistent chunks / # of chunks | # of chunk buffers |
| --- | ----------------------------------- | --------------------------------------- | -------------------- | ------------------------------------ | ------------------ |
| A   | GPT-1B BS=8 on 4 RTX 3090s or A100s | 0 / 32                                  | 0                    | 12 / 12                              | 0                  |
| B   | GPT-1B BS=64 on 4 RTX 3090s         | 24 / 32                                 | 2                    | 2 / 12                               | 3                  |
| C   | GPT-1B BS=64 on 4 A100s             | 0 / 32                                  | 0                    | 12 / 12                              | 0                  |
| D   | GPT-10B BS=8 on 4 RTX 3090s         | 48 / 48                                 | 0                    | 3 / 49                               | 46                 |
| E   | GPT-10B BS=8 on 4 A100s             | 0 / 48                                  | 0                    | 15 / 49                              | 3                  |

(to be continued...)

---

> ### Author Response · Authors · 2024-11-18
>
> We also summarized how the best configurations identified by Protrain changes under different batch sizes, hardware, and model sizes, listed as follows:
>
> **Batch Size Impact**: When increasing the batch size from 8 (row A) to 64 (row B) on RTX 3090 GPUs, the optimal configuration changes as follows: the number of swapping blocks increases from 0 to 2, the number of activation checkpointing blocks increases from 0 to 24, the number of persistent chunks decreases from 12 to 2, and the number of chunk buffers increases from 0 to 3. These configurations align with runtime execution patterns. A larger batch size increases the computation intensity of the forward and backward pass, making it possible for parameter uploads to be fully hidden by the computation.
>
> As a result, as the batch size increases, ProTrain prioritizes offloading and thus uses fewer persistent chunks and more swapping blocks to save GPU memory. ProTrain selects 2 swapping blocks as the swapping overhead for 2 blocks can be effectively overlapped with computation without impacting parameter prefetching.
>
> **Hardware Impact**: When training GPT-1B with BS=8 (row A), GPU memory is sufficient on both A100 and RTX3090 hardware. Therefore, no offloading or activation checkpointing is required, and the configurations are identical. For GPT-1B with BS=64 (rows B and C), A100 GPUs have sufficient memory, while RTX 3090 requires offloading and checkpointing, leading to different configuration choices. For GPT-10B with BS=8 (rows D and E), both hardware lack sufficient memory, but their configurations differ due to varying runtime patterns. RTX 3090s, lacking NVLink and being communication-bound for NCCL operations, use checkpointing for all blocks to allocate more space for larger chunk buffers and persistent chunks, reducing parameter gathering overhead that cannot be fully hidden by computation. In contrast, A100 GPUs, equipped with NVLink and thus have a much higher communication bandwidth, retain all activations and save memory by offloading model states, using fewer chunk buffers and persistent chunks.
>
> **Model Size Impact**: The table clearly shows that different model sizes require different configuration combinations. As the model size increases, there is generally more offloading (fewer persistent chunks and chunk buffers, more swapping blocks) and more activation checkpointing (more checkpointing blocks).

---

### Author Response · Authors · 2024-11-27

Thank you for taking the time to review our paper and provide valuable feedback. We have carefully addressed all the raised questions and provided detailed responses. As the author-reviewer discussion period is nearing its end, we kindly request any feedback you may have on our responses to ensure we have thoroughly addressed your concerns and further improved the work.

We hope that you will support our paper’s acceptance given the recognized strengths of the paper. Specifically, our work aims to make LLM training more accessible on resource-constrained GPUs, which we believe is a crucial advancement for the field. We have endeavored to address key challenges and have detailed these efforts in our submission. Thank you for your consideration and support!

---

### Meta-Review · Area_Chair_FzCV · 2024-12-20

**Metareview:**

This paper proposes a novel system for automatically tuning memory management when training LLMs. While the paper tackles an important problem, the scope of the technical contributions is limited; furthermore, there are some concerns about the scope of the experiments and the baselines considered. The authors are encouraged to improve their paper either by significantly strengthening their experiments, or by expanding on the technical contribution to achieve large performance gains (e.g., expanding the search space of optimizations considered).

**Additional Comments On Reviewer Discussion:**

The reviewers raised a variety of different concerns, but they all pointed to flaws in the experimental setup as well as the limited scope of the technical contribution. These concerns persisted after the rebuttal period.

---

### Decision · Program_Chairs · 2025-01-22

Reject